# CDKAL1 dysfunction impairs lysine codon translation in podocytes and accelerates chronic kidney disease

Hiroko Nagata [ID] [1,2], Yu Nagayoshi [1,3], Takeshi Chujo [ID] [1], Hitomi Kaneko [ID] [1], Kayo Nishiguchi [1,3], Yutaka Kakizoe [3], Hiroko Ijima [1,4], Korin Sakakida [1,5], Takeshi Masuda [6,7], Sumio Ohtsuki [7], Fan-Yan Wei [8], Yukie Takahashi [9,10], Takaichi Fukuda [9], Hideaki Jinnouchi [4], Yuki Adachi [1,11], Ryosuke Yamamura [1,3], Koki Matsushita [1,3], Masataka Adachi [3], Hideki Yokoi [3], Kimitoshi Nakamura [2], Hitoshi Nakazato [2] & Kazuhito Tomizawa [ID] [1,12] ✉

## Abstract

*Cdk5 regulatory subunit-associated protein 1-like 1 (Cdkal1)* encodes a tRNA-modifying enzyme responsible for thiomethylation generating 2-methylthio-$N^6$-threonylcarbamoyladenosine ($ms^2t^6A$) in the anticodon loop of $tRNA^{Lys}_{UUU}$. Genome-wide association studies have identified *CDKAL1* variants as risk factors for type 2 diabetes mellitus (DM) and chronic kidney disease (CKD), but whether CKD arises independently of diabetes has remained elusive. Here, we demonstrate that CDKAL1 is required for kidney function and that its dysfunction directly promotes CKD progression independently of diabetes. Systemic and podocyte-specific *Cdkal1* knockout in mice leads to CKD phenotypes in later adulthood or after increasing the burden on kidney. *Cdkal1*-knockout podocytes show reduced lysine-codon translation and decreased levels of lysine-rich proteins, including such that are important for podocyte functions, accompanied by impaired cell migration. These adverse effects on podocytes could be partially reversed by overexpressing CD2AP, a lysine-rich protein. These findings extend the concept of 'tRNA modopathy' to kidney disease and provide mechanistic insights into how defective tRNA modification contributes to kidney disease progression.

Keywords CDKAL1; Chronic Kidney Disease (CKD); tRNA Modification Enzyme; Podocyte Dysfunction; Lysine Codon Translation
Subject Categories Molecular Biology of Disease; RNA Biology; Translation & Protein Quality

## Introduction

The number of patients with end-stage kidney disease (ESKD) who require dialysis is increasing significantly worldwide (Couser et al, 2011). For early therapeutic intervention, chronic kidney disease (CKD) is defined as the preliminary stage of ESKD. CKD is diagnosed in the presence of low glomerular filtration rate (GFR) or the persistence of proteinuria, which is suggestive of kidney damage. In particular, albuminuria is one of the risk factors for CKD progression (Levin and Stevens, 2014). Hence, the early detection and treatment of patients with CKD are important for preventing progression to ESKD (Couser et al, 2011; Levin and Stevens, 2014).

The glomeruli of the kidney perform renal filtration and are composed of endothelial cells, mesangial cells, and glomerular epithelial cells (podocytes). Podocytes have large cell bodies with primary processes extending from the cell body and additional foot processes extending from the primary processes. The foot processes form frequent connections with the foot processes of neighboring glomerular podocytes to create a slit membrane (Nagata, 2016). These cells are terminally differentiated and have a limited ability to proliferate (Lasagni et al, 2013). Podocytes are the final filtration barrier, and mutations in the podocyte proteins nephrin and podocin, which are encoded by *NPHS1* and *NPHS2*, respectively, have been identified as the causes of congenital nephrotic syndrome (Boute et al, 2000; Kestilä et al, 1998; Roselli et al, 2004). In recent years, many cases of CKD with albuminuria have been attributed to disorders of glomerular podocytes (podocytopathy) (Kopp et al, 2020). However, the molecular mechanism underlying the link between CKD and podocyte function has been poorly characterized.

Transfer RNAs (tRNAs) work as adapters linking messenger RNAs (mRNAs) and amino acids. tRNAs bear various chemical modifications, which contribute to the stability of tRNA structure

[1]Department of Molecular Physiology, Faculty of Life Sciences, Kumamoto University, Kumamoto, Japan. [2]Department of Pediatrics, Faculty of Life Sciences, Kumamoto University, Kumamoto, Japan. [3]Department of Nephrology, Faculty of Life Sciences, Kumamoto University, Kumamoto, Japan. [4]Jinnouchi Hospital, Kumamoto, Japan. [5]Department of Metabolic Medicine, Faculty of Life Sciences, Kumamoto University, Kumamoto, Japan. [6]Institute for Advanced Biosciences, Keio University, Tsuruoka, Japan. [7]Department of Pharmaceutical Microbiology, Faculty of Life Sciences, Kumamoto University, Kumamoto, Japan. [8]Department of Modomics Biology and Medicine, Institute of Development, Aging and Cancer, Tohoku University, Sendai, Japan. [9]Department of Anatomy and Neurobiology, Faculty of Life Sciences, Kumamoto University, Kumamoto, Japan. [10]Division of Hematology & Oncology, Department of Internal Medicine, Iwate Medical University School of Medicine, Iwate, Japan. [11]Department of Gastroenterological Surgery, Faculty of Life Sciences, Kumamoto University, Kumamoto, Japan. [12]International Research Center for Medical Sciences, Kumamoto University, Kumamoto, Japan. ✉E-mail: tomikt@kumamoto-u.ac.jp

and the efficiency of mRNA translation (Chujo and Tomizawa, 2021; Davyt et al, 2023; Kapur and Ackerman, 2018; Suzuki, 2021). The most extensively modified nucleosides are at tRNA positions 34 and 37, which are the first nucleotide of an anticodon and the nucleotide adjacent to the anticodon, respectively. Modifications at these positions contribute to accurate and efficient decoding (Helm and Alfonzo, 2014; Kapur and Ackerman, 2018; Suzuki, 2021). In clinical research, the use of $N^1$-methylpseudouridine modification has made significant contributions to the development of mRNA vaccines against SARS-CoV-2 by improving mRNA stability and reducing the immunogenicity of the in vitro-synthesized mRNAs (Karikó et al, 2005; Karikó et al, 2008). RNAs are modified by specific enzymes, and owing to the physiologic importance of RNA modifications, pathogenic mutations have been identified in approximately 40 tRNA modification enzymes. The diseases caused by the loss of tRNA modifications are collectively referred to as "tRNA modopathies". We previously reported the pathogenic mechanisms whereby the loss of specific tRNA modifications cause various tRNA modopathies, such as in type 2 diabetes mellitus (Wei et al, 2011; Zhou et al, 2014), X chromosome-linked intellectual disability (Nagayoshi et al, 2021), and mitochondrial disease (Ahmad et al, 2024; Murakami et al, 2023). With respect to nephrology, Galloway–Mowat syndrome is a genetic disease that causes microcephaly and massive albuminuria in early childhood, and also leads to ESKD. This syndrome is caused by mutations in the *WDR4* (*WD repeat domain 4*) gene, which encodes a subunit of the tRNA guanine-$N^7$-methyltransferase (Braun et al, 2018), or mutations in the genes encoding components of the Kinase, Endopeptidase, and Other Proteins of Small size (KEOPS) complex (encoded by *YRDC, OSGEP, TPRKB, TP53RK, LAGE3,* and *GON7*), which modifies adenosine at position 37 of tRNA to $N^6$-threonylcarbamoyladenosine ($t^6A$; Fig. EV1A) (Arrondel et al, 2019; Beenstock et al, 2020; Braun et al, 2017). However, because Galloway–Mowat syndrome also involves brain abnormalities, such as microcephaly and intellectual disability, it is unclear whether the deficits in tRNA modification directly cause kidney disorders or indirectly impair kidney function *via* dysfunction in other organs.

Large genome-wide association studies have shown that SNPs in the *CDKAL1* (*Cdk5 regulatory subunit-associated protein 1-like 1*) gene are closely associated with the development of type 2 DM (Saxena et al, 2007; Scott et al, 2007; Steinthorsdottir et al, 2007; Zeggini et al, 2007). A specific SNP (rs7756992) in an intron of *CDKAL1* was reported to decrease *CDKAL1* mRNA expression (Zhou et al, 2014) and reduce insulin secretion by 22% and increase the incidence of type 2 DM up to two-fold compared with individuals at low risk (Steinthorsdottir et al, 2007). We have shown that CDKAL1 is an enzyme that thiomethylates $t^6A$ to 2-methylthio-$N^6$-threonylcarbamoyladenosine ($ms^2t^6A$) at position 37 of $tRNA^{Lys}_{UUU}$ (Fig. 1A,B) (Wei et al, 2011). This modification by CDKAL1 ensures the accurate translation of the lysine codons AAG and AAA (Naaman and Bakris, 2023). In human pre-proinsulin mRNA, the AAG codon is used to encode Lys53 and Lys88 (Narendran et al, 2020). The mutations of *CDKAL1* result in mistranslation and decreased translation at lysine codons, and therefore secretion of less mature insulin, because Lys88 is one of the protease cleavage sites that are required to cleave the protein to yield mature insulin and C-peptide (Narendran et al, 2020). Type 2 DM is the most common cause of CKD and ESKD (Naaman and Bakris, 2023). The associations between *CDKAL1* gene mutations

and CKD in patients with type 2 DM is a significant risk factor for and a predictor of CKD (Jiang et al, 2016). However, whether CKD associated with the *CDKAL1* SNP is a secondary effect of type 2 DM or is directly caused by impaired function of CDKAL1 in the kidney is unclear, despite this knowledge being pivotal for the development of measures to prevent or treat CKD in future.

In this study, we elucidated the molecular function of CDKAL1 in podocytes and the mechanism whereby CDKAL1 dysfunction promotes CKD. First, we generated systemic *Cdkal1* knockout (KO) mice and podocyte-specific *Cdkal1* KO mice. These mice showed albuminuria, impairments in kidney function, and morphologic abnormalities of their podocytes. Next, we established *Cdkal1* KO podocyte cell lines, which showed poor motility and inefficient lysine translation. Importantly, we also found decreased levels of various lysine-rich proteins that are involved in podocyte function, including the representative lysine-rich protein CD2-associated protein (CD2AP). We also performed *Cd2ap* overexpression in *Cdkal1* KO podocytes, which restored their motility. This study demonstrates that the dysfunction of a tRNA-modifying enzyme causes podocytopathy, increases the severity of albuminuria, and accelerates the progression of CKD. These findings extend the concept of 'tRNA modopathy' to nephrology and suggest avenues for research into new therapeutic strategies for CKD.

# Results

## Systemic *Cdkal1* KO mice exhibit phenotypes of CKD progression upon increased kidney load

To investigate whether dysfunction of CDKAL1 causes CKD, we first generated systemic *Cdkal1* KO mice by mating mice carrying a Cre recombinase gene ubiquitously transcribed by the CAG promoter with mice in which loxP sequences flanking *Cdkal1* exon 5 (flox mice) had been inserted (Fig. 1C). Systemic *Cdkal1* KO mice showed no significant difference in body mass from the floxed mice (Fig. 1D). Next, we performed 5/6 nephrectomy to create a model of CKD (Hashimoto et al, 2021; Wang et al, 2019). Serum and urine samples were collected from these mice and their renal (dys) function including albuminuria was evaluated. We found that the urine albumin levels in systemic *Cdkal1* KO mice were elevated than that of the floxed mice only after 5/6 nephrectomy (Fig. 1E). However, there were no differences in urinary nitrogen and sodium excretion into the urine between the two groups (Fig. EV1B,C), indicating that systemic *Cdkal1* KO mice have no impairment in kidney tubule function. In addition, we measured the serum creatinine levels in these mice, and found that they were elevated in systemic *Cdkal1* KO mice only after 5/6 nephrectomy (Fig. 1F). This suggests that kidney function is impaired in systemic *Cdkal1* KO mice subjected to this intervention. We also administered streptozotocin (STZ) to these mice to create another model of CKD (Glastras et al, 2016), which showed that the systemic *Cdkal1* KO mice had higher urine albumin than the floxed mice after STZ administration (Fig. 1G). Next, we evaluated the pathology of these mice using Periodic acid–Schiff (PAS) staining of the kidney glomeruli, which revealed no histologic abnormalities *vs.* the floxed mice (Fig. 1H). We also examined the tubules and blood vessels of the systemic *Cdkal1* KO mice after hematoxylin and eosin (HE) and Azan–Mallory staining, which revealed that they had no

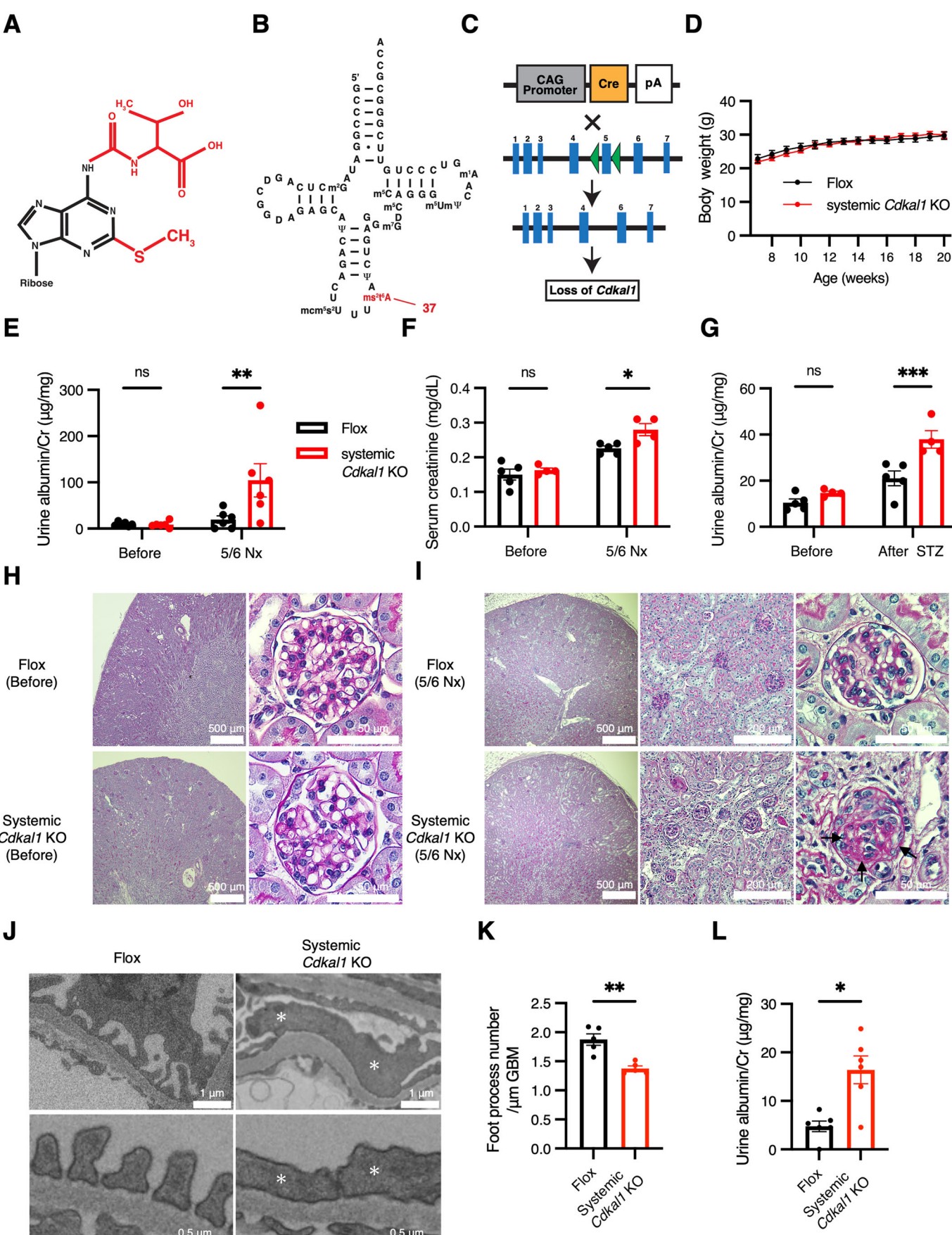

**Figure 1. CKD progression phenotypes observed upon increased kidney load in systemic *Cdkal1* KO mice.**

(A) Chemical structure of $ms^2t^6A$. The modified residues of 2-methylthio-$N^6$-threonylcarbamoyladenosine ($ms^2t^6A$) are depicted in red and the adenosine backbone in black. (B) Secondary structure of the human cytoplasmic tRNA$^{Lys}_{UUU}$, with the following modified nucleosides: $N^2$-methylguanosine ($m^2G$), dihydrouridine (D), pseudouridine (Ψ), 5-methoxycarbonylmethyl-2-thiouridine ($mcm^5s^2U$), $N^7$-methylguanosine ($m^7G$), 5-methylcytidine ($m^5C$), 5,2′-O-dimethyluridine ($m^5Um$), and $N^1$-methyladenosine ($m^1A$). (C) Generation of systemic *Cdkal1* KO mice. Mice with *Cdkal1* exon 5 flanked by loxP sequences (green triangles) were crossed with ubiquitously Cre-expressing mice to generate systemic *Cdkal1* KO mice. pA, rabbit β globin poly-adenylation signal. (D) Body masses of systemic *Cdkal1* KO mice and floxed mice. $n = 6$. Data are presented as mean ± SEM. (E) Urine albumin levels normalized by urine creatinine (Cr) before (8-week-old) and after 5/6 nephrectomy (Nx) (12-week-old). $n = 6$ each. **$P = 0.0079$. (F) Serum creatinine levels. $n = 4–5$. *$P = 0.0197$. (G) Urine albumin level normalized by urine creatinine before (8-week-old) or after streptozotocin (STZ) administrations (33-week-old). $n = 4–5$. ***$P = 0.0009$ by two-way ANOVA followed by Sidak post-hoc test (E–G). Data are presented as mean ± SEM. (H) Representative images of the kidney glomeruli of systemic *Cdkal1* KO mice and floxed mice (10-week-old). Scale bars, 500 μm (left), 50 μm (right). (I) Representative images of the kidney glomeruli of systemic *Cdkal1* KO mice after 5/6 nephrectomy (12-week-old). Arrows indicate areas of segmental sclerosis within the glomeruli. Scale bars, 500 μm (left), 200 μm (center), and 50 μm (right). (J) Representative scanning electron microscope images of 40-week-old systemic *Cdkal1* KO or floxed mouse glomeruli. Asterisks indicate foot process effacement. Scale bars, 1 μm (upper panels), 0.5 μm (lower panels). (K) Quantitative analysis of foot process number/μm glomerular basement membrane in systemic *Cdkal1* KO mice. Data were obtained from five glomeruli of one floxed mouse and one KO mouse. $n = 5$. Data are presented as mean ± SEM. **$P = 0.079$ by Mann–Whitney $U$ test. (L) Urine albumin levels normalized by urine creatinine (Cr) 50-week-old mice. $n = 6$. Data are presented as mean ± SEM. *$P = 0.0152$ by Mann–Whitney $U$ test. Source data are available online for this figure.

morphologic abnormalities (Fig. EV2A,B). However, after 5/6 nephrectomy, the glomeruli of systemic *Cdkal1* KO mice showed focal segmental glomerulosclerosis (FSGS)–like features. Sclerotic changes with extracellular matrix deposition were observed within the affected glomeruli (Fig. 1I, arrows). These results suggest that CKD progresses in systemic *Cdkal1* KO mice upon increased kidney loads, likely *via* glomerular injury and not tubular damage.

## Systemic *Cdkal1* KO mouse podocytes show morphologic abnormalities

Next, we investigated the localization of CDKAL1 in the mouse glomeruli by immunostaining. Using a CDKAL1 antibody that generated immunofluorescence signals in the kidneys of floxed mice, but not in systemic *Cdkal1* KO mice (Fig. EV3A), we found that CDKAL1 protein was expressed in podocytes by co-staining with podocin, a podocyte-specific protein (Fig. EV3B). CDKAL1 was also expressed in mesangial and endothelial cells (Fig. EV3B). Thus, CDKAL1 was present in all the cells of the mouse glomeruli. We next investigated whether the loss of CDKAL1 injured the glomeruli by examining them using transmission electron microscope. The effacement of foot processes was observed in the glomeruli of the systemic *Cdkal1* KO mice at 40 weeks of age, but not in control floxed mice of the same age (Fig. 1J). Consistently, quantitative analysis revealed that podocytes in systemic *Cdkal1* KO mice had significantly fewer foot processes per micrometer of the glomerular basement membrane (GBM) (Martin et al, 2022) (Fig. 1K). Functionally, a mild increase in albuminuria was also detected in systemic *Cdkal1* KO mice at 40 weeks (Fig. 1L), whereas PAS staining revealed no tubular abnormalities (Fig. EV3C), indicating that albuminuria originated from glomerular injury. By contrast, in the systemic *Cdkal1* KO mice at 8 weeks of age, foot process effacement was not observed (Appendix Fig. S1A,B). Therefore, systemic *Cdkal1* KO mice show podocyte abnormalities in later adulthood.

## Podocyte-specific *Cdkal1* KO mice also exhibit the phenotypes of CKD progression upon increased kidney load

In the experiments described above, systemic *Cdkal1* KO mice showed kidney dysfunction and podocyte abnormalities. To investigate whether these effects in the kidney occurred independent of DM, we next generated podocyte-specific *Cdkal1* KO mice by mating floxed mice with mice carrying a Cre recombinase gene driven by the promoter of the podocin-encoding *Nphs2* gene, to create mice with podocyte-specific Cre expression (Moeller et al, 2003). The body mass of podocyte-specific *Cdkal1* KO mice was not significantly different from that of floxed mice (Fig. 2A). Patients carrying variants in the *CDKAL1* gene exhibit specific impairments in first-phase insulin secretion (Groenewoud et al, 2008). Consistently, pancreatic β-cell-specific *Cdkal1* KO mice display impaired glucose tolerance compared with floxed mice, and plasma insulin levels measured 15 min after glucose loading are significantly reduced in β-cell KO mice (Wei et al, 2011). To verify that pancreatic function and insulin production were not compromised in podocyte-specific *Cdkal1* KO mice and to ensure that renal dysfunction was not secondary to defects in other organs, we performed an intraperitoneal glucose tolerance test and monitored insulin secretion during the test. Podocyte-specific *Cdkal1* KO mice exhibited normal glucose tolerance and insulin secretion, indicating that pancreatic function and insulin production were unaffected (Fig. 2B,C). In the glomeruli of podocyte-specific *Cdkal1* KO mice, immunostaining showed no colocalization of CDKAL1 with the podocyte-specific proteins nephrin and synaptopodin (Appendix Fig. S2A). Next, we performed PAS staining, and found no morphologic abnormalities in the kidney glomeruli of the podocyte-specific *Cdkal1* KO mice (Fig. 2D). HE and Azan–Mallory staining also showed no morphologic abnormalities in the tubules or blood vessels of the podocyte-specific *Cdkal1* KO mice (Fig. EV2A,B). We next performed 5/6 nephrectomy of podocyte-specific *Cdkal1* KO mice and collected serum and urine samples to evaluate their kidney function. As a result, we observed elevations of urine albumin in the podocyte-specific *Cdkal1* KO mice than in floxed mice, only after 5/6 nephrectomy (Fig. 2E). We evaluated tubular function by measuring urinary nitrogen and sodium excretion, and found no difference between the podocyte-specific *Cdkal1* KO mice and floxed mice (Fig. EV4A,B). Importantly, after 5/6 nephrectomy, the serum creatinine of the podocyte-specific *Cdkal1* KO mice was elevated (Fig. 2F). We also identified FSGS features in the glomeruli of the podocyte-specific *Cdkal1* KO mice (Fig. 2G, arrows). Furthermore, STZ administration also resulted in higher urine albumin levels in the podocyte-specific *Cdkal1* KO mice than those in the floxed

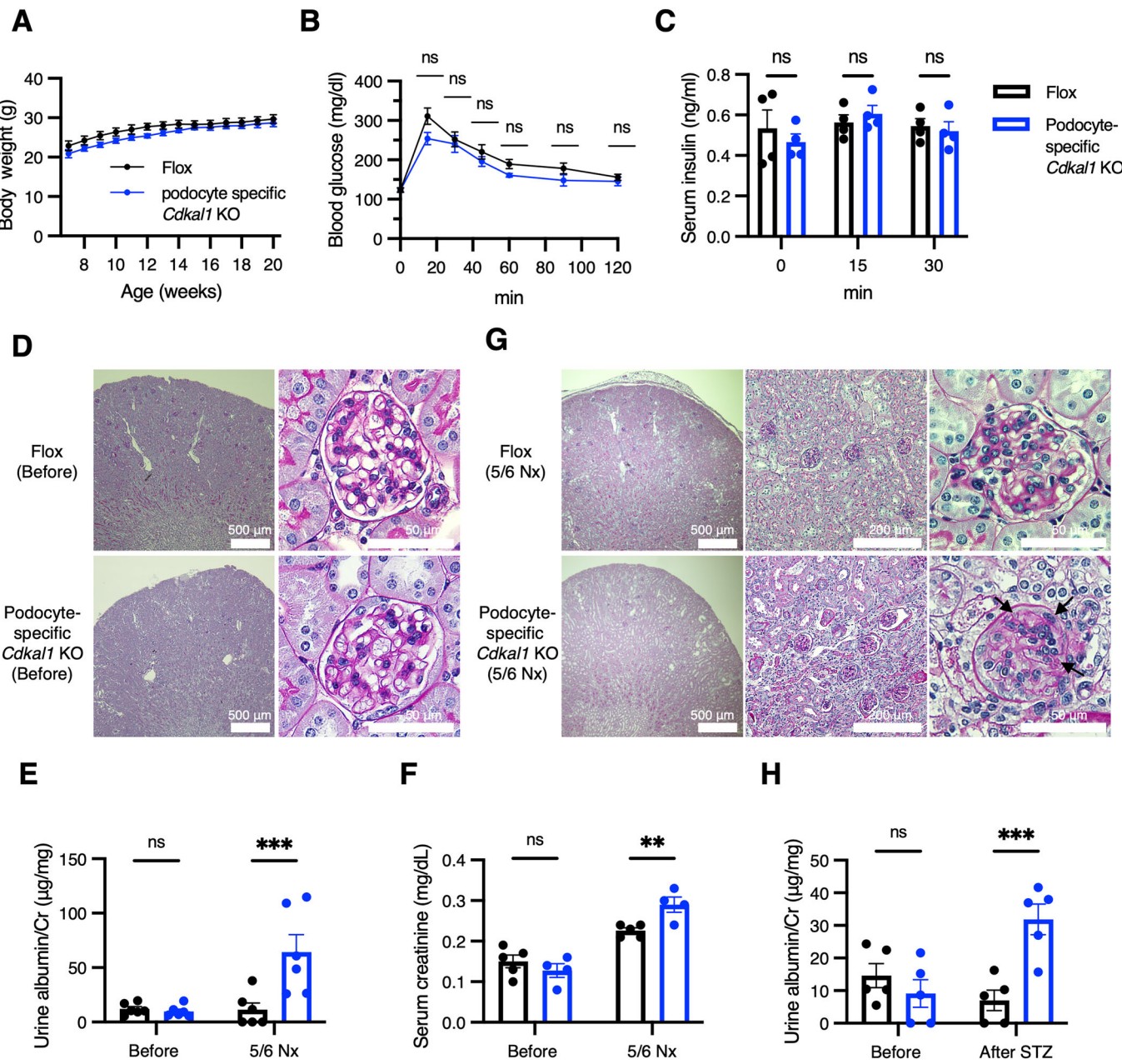

**Figure 2.  CKD progression phenotype observed upon increased kidney load in podocyte-specific *Cdkal1* KO mice.**

(A) Body masses of $n = 6$ podocyte-specific *Cdkal1* KO mice and floxed mice. Data are presented as mean ± SEM. (B, C) Blood glucose (B) and serum insulin levels (C) during glucose tolerance test at 20 weeks. $n = 6$ for glucose and $n = 4$ for insulin. Data are presented as mean ± SEM; n.s., not significant by Mann–Whitney $U$ test. (D) PAS-stained kidney glomeruli from podocyte-specific *Cdkal1* KO and floxed mice at 10 weeks of age. Scale bars, 500 µm (left), 50 µm (right). (E) Urine albumin level normalized by urine creatinine (Cr) before (8-week-old) or after 5/6 nephrectomy (Nx) (16-week-old). $n = 6$ each. ***$P = 0.0007$. (F) Serum creatinine levels. $n = 4$–5. **$P = 0.0085$. (G) PAS-stained kidney glomeruli of podocyte-specific *Cdkal1* KO mice after 5/6 nephrectomy (16-week-old). Arrows indicate areas of segmental sclerosis within the glomeruli. Scale bars, 500 µm (left), 200 µm (center), and 50 µm (right). (H) Urine albumin level normalized by urine Cr in $n = 5$ podocyte-specific *Cdkal1* KO mice and flox mice before (8-week-old) or after STZ administrations (28-week-old). ***$P = 0.0009$. (E, F, H): mean ± SEM, two-way ANOVA followed by Sidak post-hoc test. Source data are available online for this figure.

mice (Fig. 2H). Interestingly, the phenotype of these mice was largely consistent with that of the systemic *Cdkal1* KO mice (Fig. 1D–I). These results collectively demonstrate that a defect in CDKAL1 in podocytes directly promotes the progression of CKD, demonstrated by albuminuria and renal dysfunction, upon kidney loading.

## *Cdkal1* KO reduces the efficiency of lysine translation, and impairs the glomerular filtration barrier and the motility of podocyte cell lines

To elucidate the mechanisms explaining how the loss of *Cdkal1* impairs podocyte function, we knocked out *Cdkal1* using the

CRISPR–Cas9 system in the immortalized mouse podocyte cell lines E11 and SVI (Appendix Figs. S3 and S4A). The loss of CDKAL1 in the KO cells was strongly suggested by the absence of ms²t⁶A modification in total RNA of $Cdkal1$ KO podocytes, assessed using liquid chromatography–mass spectrometry (LC–MS) (Fig. 3A; Appendix Fig. S4B, and Reagents and Tools Table). Next, because CDKAL1 specifically modifies $tRNA^{Lys}_{UUU}$, we evaluated the translation efficiency of lysine codons. To this end, we used codon translation reporter plasmids, which contain five consecutive codons of interest at the 5′ end of the $renilla$ $luciferase$-coding sequence, as well as a $firefly$ $luciferase$ gene as a loading control (Fig. 3B). We transfected the control cells or $Cdkal1$ KO E11 cells with the codon translation reporter plasmids (random codon control, Lys AAA codon, Lys AAG codon, or control Phe TTT codon) and compared the lysine codon translation efficiency of the cells. We found that $Cdkal1$ KO E11 cells showed reduction in codon translation efficiency respect to both of the two lysine codons, AAA and AAG (Fig. 3C). As a negative control, we confirmed that the codon translation efficiency of phenylalanine TTT, which is unrelated to the CDKAL1 substrate $tRNA^{Lys}_{UUU}$, was unaffected by $Cdkal1$ KO in E11 cells.

Podocytes play a crucial role in maintaining the glomerular filtration barrier by regulating the selective passage of solutes and preventing the loss of macromolecules such as albumin (Nagata, 2016). Disruption of podocyte integrity leads to impaired glomerular filtration barrier function. Therefore, we assessed glomerular filtration barrier function using $Cdkal1$ KO cell lines. Compared to sgControl cells, $Cdkal1$ KO cells exhibited increased albumin influx across the podocyte monolayer, indicating a compromised filtration barrier (Fig. 3D). The motility of podocytes is needed for the formation, maintenance, and dynamic remodeling of foot processes, which are particularly important for the response to injury (Falkenberg et al, 2017; Tian and Ishibe, 2016). Loss of this motility leads to the effacement of foot processes, podocyte shedding, inflammation, and fibrosis, and this promotes the progression of CKD (Reynolds, 2020). To evaluate the motility of the $Cdkal1$ KO podocyte cell line, we first performed a wound healing migration assay, and found an impairment of migration ability in the $Cdkal1$ KO E11 cells (Fig. 3E,F) and SVI cells (Fig. 3G,H). We also evaluated the intracellular distribution of filamentous actin (F-actin) cytoskeleton in $Cdkal1$ KO E11 and SVI cells by immunostaining, and found that the cytoskeletal network was disrupted and diminished in the $Cdkal1$ KO podocyte cell lines (Fig. 3I–K). Notably, CDKAL1 KO in a human hepatocarcinoma cell line (HuH-7) did not alter F-actin organization (Appendix Fig. S5A–C), indicating the tissue/cell-type-specific importance of CDKAL1 for F-actin formation. Collectively, these results indicated that a defect in $Cdkal1$ causes an abnormality in F-actin polymerization, which negatively affects podocyte motility.

To elucidate the potential mechanisms that connect impaired lysine translation with the dysfunction of podocytes, we performed proteomic analysis to compare protein expression in the $Cdkal1$ KO E11 and control cells. We selected proteins that had at least four-fold higher or lower expression in $Cdkal1$ KO E11 cells $vs$. control cells, which revealed that 48 and 148 proteins were expressed at higher and lower levels, respectively (Fig. 4A). We next performed gene ontology (GO) analysis using the genes corresponding to these proteins. The enriched molecular function GO categories included "extracellular matrix structural constituent" and "integrin binding"

(Fig. 4B). The enriched cellular component GO categories included "glomerular basement membrane", "extracellular matrix", "lamelli-podium", and "glomerular filopodium" (Fig. 4C). Thus, the results of the GO analysis of the genes corresponding to the 196 proteins that were differentially expressed in the $Cdkal1$ KO podocytes are highly consistent with the impaired migration phenotype.

We next evaluated the lysine ratios of these proteins. The proteins that decreased in $Cdkal1$ KO podocytes had higher lysine ratios than the increased proteins (Fig. 4D). The ms²t⁶A modification at position 37 of $tRNA^{Lys}_{UUU}$ has been reported to enhance the binding of $tRNA^{Lys}_{UUU}$ to AAA and AAG codons (Narendran et al, 2020; Yarian et al, 2000). Usages of both AAA and AAG codon were higher in proteins that decreased in $Cdkal1$ KO podocytes than in the increased proteins (Fig. 4E,F). This result is consistent with impaired translation efficiency of both AAA and AAG codons in $Cdkal1$ KO podocytes (Fig. 3C). The expression of the mRNAs encoding these proteins, assessed using RNA-seq, did not differ between the $Cdkal1$ KO podocytes and control cells (Appendix Fig. S6A). Collectively, these results demonstrate that a defect in CDKAL1 reduces translation efficiency at both lysine codons, AAA and AAG, and affects the amount of lysine-rich proteins.

## The expression of the representative lysine-rich protein CD2AP is low in the $Cdkal1$ KO podocytes and glomeruli of systemic $Cdkal1$ KO mice

To further interrogate the association between impaired lysine translation efficiency and podocyte dysfunction, we evaluated the lysine percentages of podocyte-related proteins. We calculated the lysine percentages of the proteins encoded by 26 genes for which pathogenic mutations have been described in patients with nephrotic syndrome in humans (Bierzynska et al, 2014; Trautmann et al, 2015). Among these proteins, we found that CD2AP had the highest lysine ratio of >12% (Fig. 4G). We performed western blotting of proteins extracted from $Cdkal1$ KO E11 (Fig. 4H,I) and SVI cells (Fig. 4K,L), and found that CD2AP protein was reduced in both KO cell lines. By contrast, reverse transcription–quantitative PCR (RT-qPCR) showed that the mRNA expression of $Cd2ap$ was high in the $Cdkal1$ KO E11 and SVI cells (Fig. 4J; Appendix Fig. S6B). Similarly, western blotting of mouse glomeruli harvested by the sieving of kidneys (Wang et al, 2019) showed that CD2AP protein was decreased in the glomeruli of systemic $Cdkal1$ KO mice (Fig. 4M,N). In addition, the $Cd2ap$ mRNA levels in glomeruli of the systemic $Cdkal1$ KO mice was also increased (Appendix Fig. S6C). These results suggest that the reduction in lysine translation efficiency induced by CDKAL1 deficiency decreases CD2AP protein level and induces a compensatory upregulation of $Cd2ap$ mRNA. To further assess whether this reaction is specific to CD2AP or shared by other lysine-rich proteins, we examined additional lysine-rich proteins (anillin and podocin) by using western blot in $Cdkal1$ KO podocyte (Appendix Fig. S6D–F). Consistent with the results for CD2AP, anillin and podocin showed reduced protein levels in $Cdkal1$ KO cells.

ER stress is closely associated with the progression of CKD (Cybulsky, 2013; Inagi, 2009). Regarding CDKAL1, pancreatic β cells of β-cell-specific $Cdkal1$ KO mice showed increase in spliced $Xbp1$ mRNA level, a marker of ER stress (Wei et al, 2011). Therefore, we measured the expression of ER stress marker genes to determine whether $Cdkal1$ deficiency induces an ER stress response

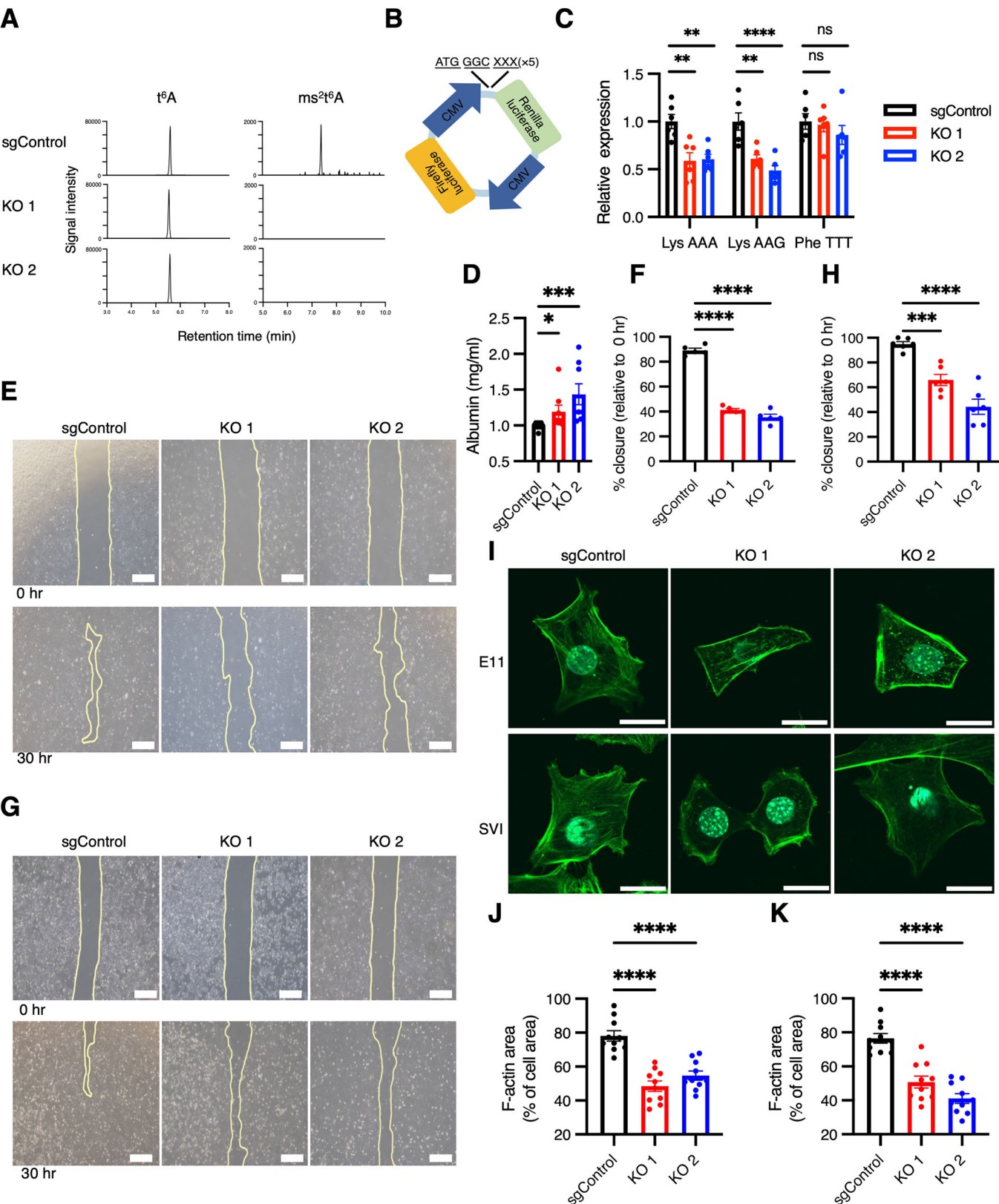

**Figure 3.** ***Cdkal1* KO leads to impaired lysine codon translation, filtration barrier dysfunction, and podocyte motility defect.**

(A) Lack of ms$^2$t$^6$A modification within total RNA of *Cdkal1* KO podocyte cells, demonstrated using LC–MS. (B) Schematic of the codon translation reporter plasmid for use in a dual-luciferase system. (C) Codon translation efficiency of *Cdkal1* KO and control E11 podocyte cells. $n = 6$ each. Data are presented as the mean ± SEM. **$P = 0.0019$ (Lys AAA sgControl vs KO1); **$P = 0.0026$ (sgControl vs KO2); **$P = 0.0013$ (Lys AAG sgControl vs KO1) and ****$P = 9.40 \times 10^{-5}$ (sgControl vs KO2) by one-way ANOVA, followed by Dunnett's correction. (D) Albumin measurements in permeability assays using *Cdkal1* knockout and control E11 podocyte cells. $n = 8$ each. Data are presented as the mean ± SEM. *$P = 0.0378$; ***$P = 0.0002$ by Kruskal–Wallis test, followed by Dunn's multiple comparison test. (E–H) Results of the wound cell migration assay in E11 podocyte cells, ****$P = 6.40 \times 10^{-10}$ (sgControl vs KO1); ****$P = 1.70 \times 10^{-10}$ (sgControl vs KO2), (E, F) and SVI podocyte cells ***$P = 0.0007$; ****$P = 1.75 \times 10^{-6}$ (G, H). Upper panels in (E) and (G): representative images of the scratched areas, created using a pipette tip, of confluent podocytes at 0 h. Lower panels in (E) and (G): representative images of the scratched areas 30 h later, after podocytes had migrated into the wounds. Scale bars, 500 µm. (F, H); $n = 5$; data are presented as the mean ± SEM, one-way ANOVA, followed by Dunnett's correction. (I) Representative images of F-actin immunostained *Cdkal1* KO E11 (upper panel) and SVI (lower panel) cells (in green). Scale bar, 20 µm. (J, K) Quantification of the cellular area occupied by F-actin in E11 podocyte cells, ****$P = 2.58 \times 10^{-7}$ (sgControl vs KO1); ****$P = 1.25 \times 10^{-5}$ (sgControl vs KO2) (J), and SVI podocyte cells, ****$P = 5.51 \times 10^{-6}$ (sgControl vs KO1); ****$P = 2.03 \times 10^{-8}$ (sgControl vs KO2) (K). $n = 10$ each. Data are presented as the mean ± SEM. Analysis by one-way ANOVA, followed by Dunnett's correction. Source data are available online for this figure.

in podocytes, and found no significant differences in the expression of these genes in *Cdkal1* KO E11 cells (Appendix Fig. S7). These results suggest that the loss of *Cdkal1* is not associated with ER stress in podocyte cell lines.

### *Cd2ap* overexpression partially restores the glomerular filtration barrier and migratory capacity in *Cdkal1* KO podocytes

Considering the reduction of CD2AP protein levels in *Cdkal1* KO cells/tissues and the importance of CD2AP for kidney function, we next investigated whether the overexpression of *Cd2ap* in *Cdkal1* KO podocytes would ameliorate the phenotype of podocytes. A lentivirus vector that induced constitutive *Cd2ap* expression was used to transduce *Cdkal1* KO SVI cells, and a resulting elevation of CD2AP protein was confirmed by western blotting (Fig. 5A). We assessed the filtration barrier function and found that *Cd2ap* overexpression in *Cdkal1* KO cells restored the filtration barrier function in KO1 cells (Fig. 5B). We then examined the cytoskeleton by immunostaining for F-actin. The overexpression of *Cd2ap* ameliorated the cytoskeletal abnormalities of the *Cdkal1* KO SVI cells, such that an orderly podocyte cytoskeleton was reinstated (Fig. 5C,D). Moreover, a wound cell migration assay revealed that *Cdkal1* KO SVI cells overexpressing *Cd2ap* showed a restoration of cell migration (Fig. 5E–G). These results indicate that low CD2AP expression can at least partially explain the dysfunction of filtration barrier, the morphologic abnormalities and impaired motility of *Cdkal1* KO SVI cells, and also that exogenous *Cd2ap* overexpression can at least partially rescue podocyte function.

## Discussion

CKD is well known to be a progressive and irreversible disease. Therefore, effective treatments based on a thorough knowledge of its molecular mechanisms are required. In the present study, the systemic *Cdkal1* KO mice showed albuminuria and kidney failure when subjected to functional overload (Fig. 1E–G), likely in the absence of abnormal reabsorption by kidney tubules (Fig. EV1B,C). Histologically, there was no effect of systemic *Cdkal1* KO on the kidneys of the mice under normal conditions, but increasing the load on the kidneys induced abnormalities in their glomeruli (Fig. 1H,I). Although CDKAL1 was found to be expressed in all components of the glomeruli (Fig. EV3B), systemic *Cdkal1* KO

mice showed podocyte effacement in later adulthood (Fig. 1J,K). These results suggest that *Cdkal1* deficiency causes abnormalities that are specific to, or most marked in, podocytes, and causes kidney dysfunction under conditions of increased kidney load. Importantly, podocyte-specific *Cdkal1* KO mice also exhibited accelerated progression of CKD and FSGS–like features when the renal load was increased, and this phenotype was largely consistent with that of the systemic *Cdkal1* KO mice (Fig. 2). Therefore, *Cdkal1* deficiency directly promotes podocyte dysfunction and accelerates the progression of CKD. To determine the molecular mechanisms whereby a loss of CDKAL1-mediated tRNA thiomethylation might cause podocyte dysfunction, we generated two *Cdkal1* KO podocyte cell lines. Because CDKAL1 modifies the nucleotide next to the anticodon of tRNA$^{Lys}_{UUU}$ alone and is important for decoding by tRNA$^{Lys}_{UUU}$ (Kaufman, 2011; Wei et al, 2011), we evaluated the translational efficiency of the lysine codons and identified reduction of translational efficiency of the lysine codons AAA and AAG (Fig. 3C). Both *Cdkal1* KO podocyte lines showed impaired barrier function and lower motility, indicating podocyte dysfunction (Fig. 3D–H). Moreover, the proteomic analysis suggested that proteins containing a high percentage of lysine residues showed lower protein levels in *Cdkal1* KO podocytes (Fig. 4D–F). Together, these findings indicate that *Cdkal1* deficiency reduces the translational efficiency of lysine codons and changes protein levels of *Cdkal1* KO podocytes.

Analysis of the amino acid compositions of the 148 decreased proteins and 48 increased proteins in *Cdkal1* KO E11 cells (Fig. 4) revealed that the decreased proteins in *Cdkal1* KO cells contained a higher ratio of Lys AAA/AAG (Fig. 4D–F). Moreover, lysine ratio correlated with the decrease in protein levels upon *Cdkal1* KO more than any other amino acids (Appendix Figs. S8 and S9). However, we also observed decrease of proteins with higher rates of amino acids encoded by near cognate codons to Lys AAA/AAG codons, such as Glu GAA/GAG (Appendix Figs. S8 and S9). This might be due to the loss of the ms$^2$ modification at position 37 of tRNA$^{Lys}_{UUU}$; tRNA$^{Lys}_{UUU}$ might show reduced codon recognition ability and might more promiscuously bind to some of the near-cognate codons, other than the cognate Lys AAA/AAG codons. However, some codons cannot be explained by this simple hypothesis, including Asp GAU/GAC (% of which increased in the decreased proteins upon *Cdkal1* KO) and Pro CCA/CCU/CCG/CCC (% of which decreased in the increased proteins upon *Cdkal1* KO). Given that altered codon recognition alone may not fully explain the reduced abundance of lysine-rich proteins, we

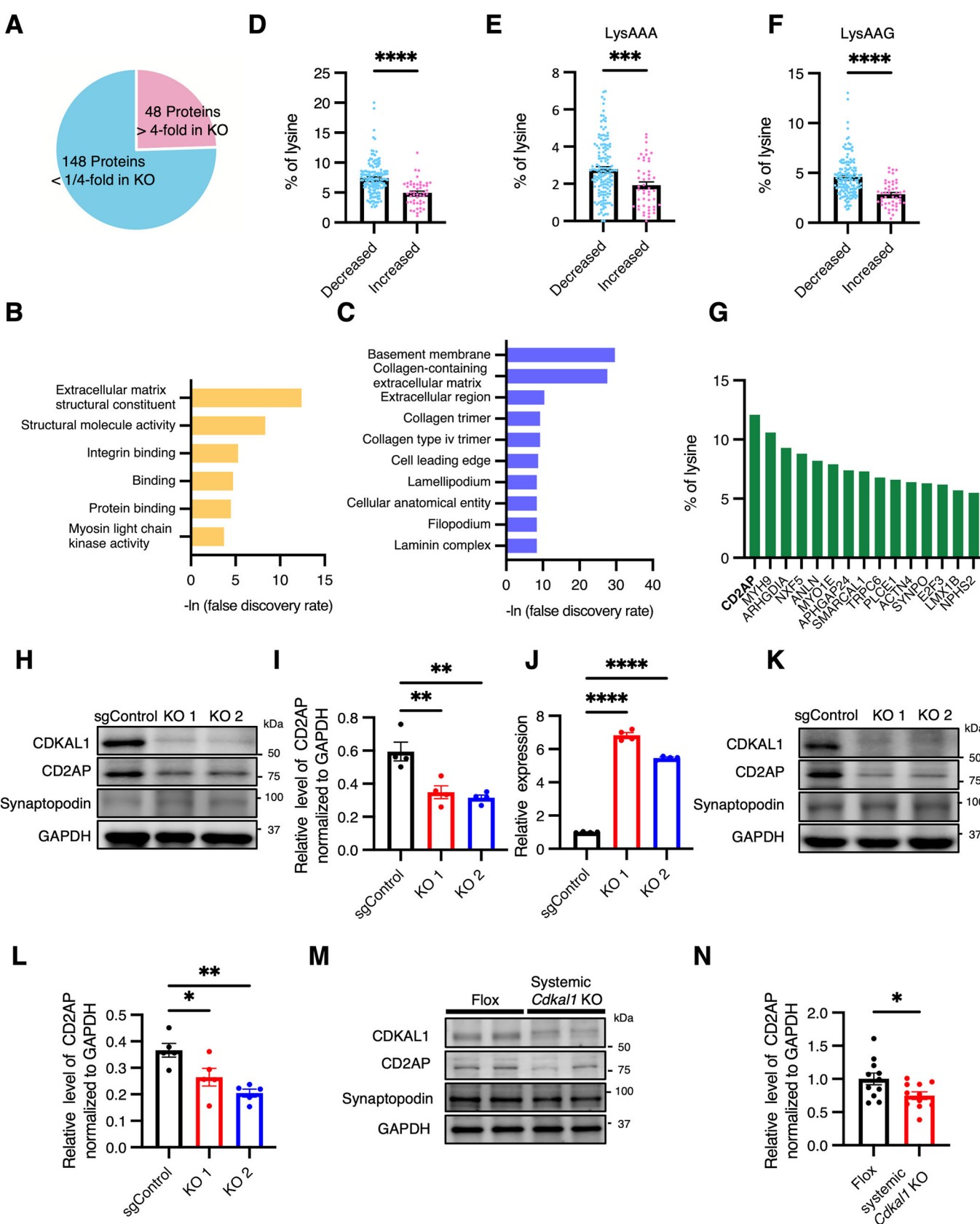

**Figure 4. Higher lysine ratio in proteins that decreased in *Cdkal1* KO podocyte cell lines, and reduction of lysine-rich protein CD2AP in *Cdkal1* KO podocytes.**

(A) Pie chart of the number of proteins with a 4-fold or greater increase or decrease of less than 1/4 in protein levels in *Cdkal1* KO E11 podocyte cells compared to Control cells. (B, C) Molecular functions (B) and cellular components (C) associated with the proteins in (A), according to Gene Ontology. (D) Lysine ratio of each protein in (A). (E, F) Percentages of lysine codons AAA (E) and AAG (F) in each protein in (D). Data are presented as the mean ± SEM. (D) ****$P = 3.39 \times 10^{-9}$; (E) ***$P = 0.0005$; (F) ****$P = 1.60 \times 10^{-9}$ by Mann–Whitney $U$ test. (G) Top 15 proteins with high lysine ratio, having human nephrotic syndrome pathogenic mutations. (H) Representative western blot images for CD2AP in *Cdkal1* KO and control E11 podocyte cells. (I, J) CD2AP protein levels normalized to GAPDH (I) and *Cd2ap* mRNA levels (J) in *Cdkal1* KO and control E11 podocyte cells. $n = 4$ each. Data are presented as the mean ± SEM. (I) **$P = 0.0040$ (sgControl vs KO1) and **$P = 0.0017$ (sgControl vs KO2). (J) ****$P = 1.99 \times 10^{-11}$ (sgControl vs KO1) and ****$P = 2.20 \times 10^{-10}$ (sgControl vs KO2). Statistical analysis by one-way ANOVA followed by Dunnett's correction. (K) Representative western blot images for CD2AP in *Cdkal1* KO and control SVI podocyte cells. (L) Quantification of CD2AP protein levels normalized to GAPDH in *Cdkal1* KO and control SVI podocyte cells. $n = 5$ each. Data are presented as the mean ± SEM. *$P = 0.029$ (sgControl vs KO1) and **$P = 0.0015$ (sgControl vs KO2) by one-way ANOVA, followed by Dunnett's correction. (M) Representative western blot images of the CD2AP of sieved glomeruli from systemic *Cdkal1* KO mice. (N) Quantification of CD2AP protein levels normalized to GAPDH. $n = 11$ each. Data are presented as the mean ± SEM; *$P = 0.040$ by Mann–Whitney U test. Source data are available online for this figure.

investigated whether enhanced degradation contributed to this phenotype. To try and address the possibility that the reduced abundance of lysine-rich proteins results from enhanced degradation via ubiquitination, we examined global ubiquitination levels in *Cdkal1* KO podocytes (Appendix Fig. S10). Western blot analysis for ubiquitin revealed no detectable differences between *Cdkal1* KO podocytes and control cells, indicating that proteins are not globally subjected to increased ubiquitin-mediated degradation. We also performed ribosome profiling to assess A-site codon occupancy (Appendix Fig. S11). We initially expected that ribosomes would slow down at Lys AAA/AAG codons in *Cdkal1* KO cells. Contrary to our initial hypothesis, the ribosomes did not show increased occupancy at Lys AAA/AAG codons (shown as red bars in Appendix Fig. S11). Nevertheless, the lysine codon reporter experiments clearly demonstrated that lysine translation was decreased in *Cdkal1* KO mice (Fig. 3C). One possible explanation for this discrepancy is that tRNAs that decode near-cognate codons may be misincorporated into the ribosomal A-site for lysine AAA/AAG codons. However, investigating this possibility requires extensive experiments, such as the quantification of tRNAs contained in translating ribosomes, and may be a good subject for future studies.

By calculating the lysine codon use in proteins that are essential for podocyte function, we found that CD2AP has the highest lysine ratio of >12% (Fig. 4G). CD2AP was originally identified as an adapter protein for the T-cell adhesion protein CD2, which is required for the formation of actin-based immunologic synapses (Dustin et al, 1998). The glomeruli of *Cd2ap* KO mice have been reported to show mesangial cell proliferation with extracellular matrix deposition, glomerulosclerosis, and foot process effacement (Shih et al, 1999). Patients who are heterozygous for the *CD2AP* variant present with proteinuria and have FSGS (Gigante et al, 2009; Liu et al, 2021). This implies that in the kidneys of the 5/6-nephrectomized systemic *Cdkal1* KO mice and podocyte-specific *Cdkal1* KO mice, kidney dysfunction (Figs. 1I and 2E) might be caused, at least in part, by reduction of CD2AP. CD2AP binds directly to α-actinin-4 and provides capping proteins for the barbed ends of polymerizing F-actin (Akin and Mullins, 2008; Maekawa and Inagi, 2017). The capping of growing filaments can increase the size of the G-actin pool that is available for branch formation and promote actin branch formation (Akin and Mullins, 2008), thereby changing the network structure and improving motility (Tang and Brieher, 2013; Zhao et al, 2013). We found that CD2AP was decreased in *Cdka1* KO podocytes and in the glomeruli of systemic

*Cdkal1* KO mice, whereas *Cd2ap* mRNA was upregulated in compensation (Fig. 4J; Appendix Fig. S6B,C). We found that *Cd2ap* overexpression restored filtration barrier integrity and the intracellular F-actin cytoskeleton stability in podocytes (Fig. 5C,D). Furthermore, the ability to migrate was restored upon overexpression of *Cd2ap* (Fig. 5E–G). CD2AP cooperates with the slit diaphragm proteins nephrin and podocin to regulate phosphoinositide 3-kinase/AKT signaling, which controls actin-based skeletal dynamics in mouse podocytes (Huber et al, 2003). Thus, the overexpression of *Cd2ap* restores the migration of *Cdkal1* KO podocytes, presumably *via* the regulation of actin.

Regarding the CKD that is associated with type 2 DM, it has been reported that SNPs in the *G6PC2* and *CDKAL1* genes increase the risk of CKD progression (Jiang et al, 2016). The study also showed that not only patients with type 2 DM and homozygosity with respect to the risk alleles of *CDKALI*, but also those that are heterozygous, are at a higher risk of CKD than those without these alleles (Jiang et al, 2016). In contrast, the risk of type 2 DM is high only in the presence of homozygosity for the risk alleles of *CDKAL1* (Steinthorsdottir et al, 2007), implying that the molecular mechanisms involved in the induction of type 2 DM and CKD might differ. A previous study showed that individuals carrying specific SNPs in *CDKAL1* (an rs7756992 A/G or G/G polymorphism) are at a 1.15-fold higher risk of progression to CKD for each G allele (Jiang et al, 2016).

Our present study demonstrates that dysfunction of CDKAL1 directly accelerates the progression of chronic kidney disease in mice. To investigate possible association in humans, we conducted a preliminary observational study. Given the exploratory nature of the study and the limited cohort size, relevant clinical data are presented in the Appendix Table S1. We investigated the association between the *CDKAL1* rs7756992 variant and CKD progression in type 2 DM patients. Participants were divided into two groups: those with the *CDKAL1* rs7756992 risk allele (+) and those without (−) (Fig. EV5A). Notably, patients with the risk allele had significantly higher levels of albuminuria (Fig. EV5B), despite having similar HbA1c, fasting insulin, and C-peptide levels to those in non-risk group (Fig. EV5C,D; Appendix Table S1). This suggests that CDKAL1 dysfunction may promote albuminuria through mechanisms independent of impaired insulin secretion. We previously reported that *Cdkal1* KO mice and humans with risk alleles show lower insulin secretion, because of reduced translation and enhanced mistranslation of insulin (Wei et al, 2011; Xie et al, 2013). By contrast, in the present study, we found no differences in

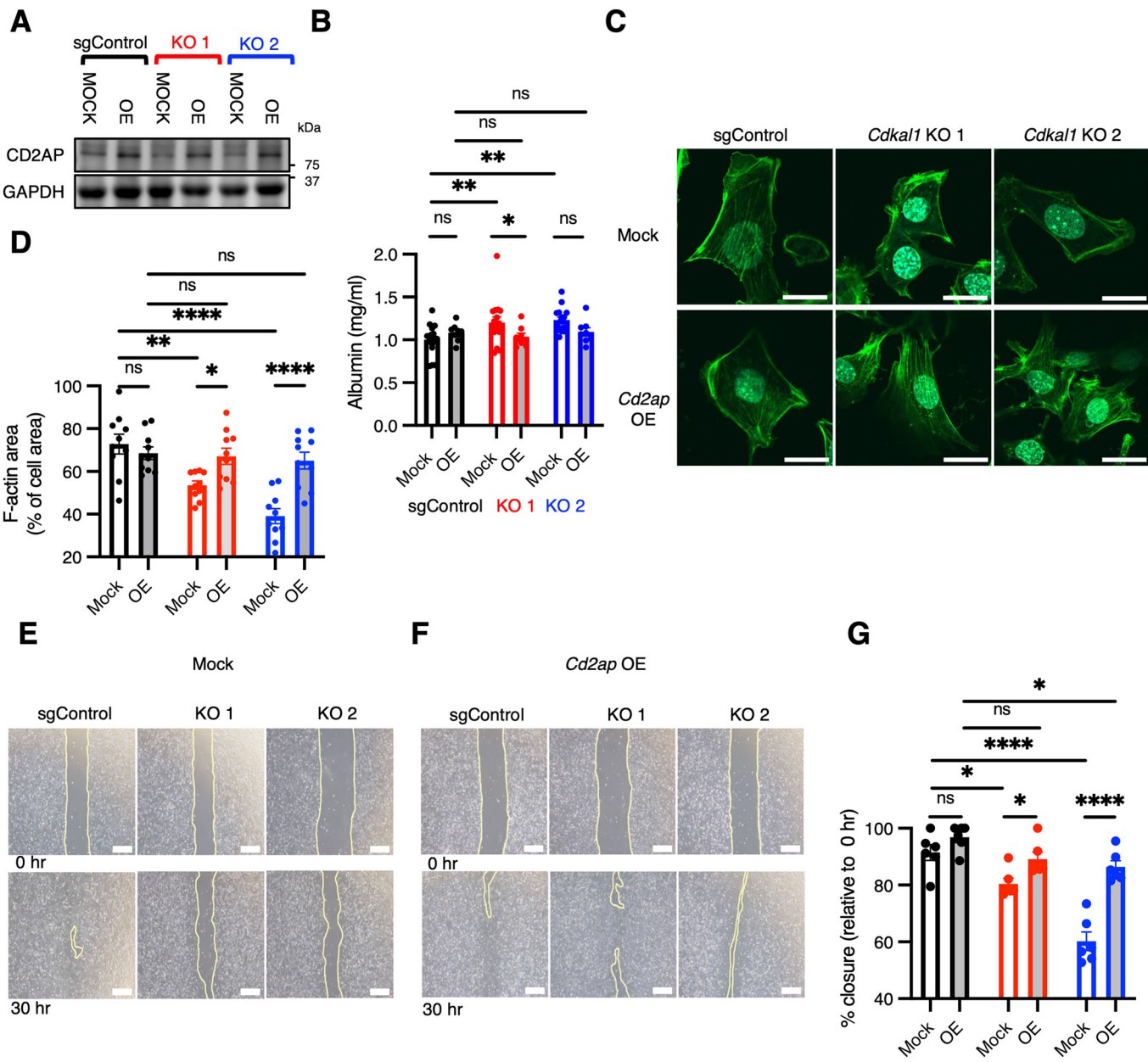

**Figure 5. *Cd2ap* overexpression partially restores filtration barrier function and migration in *Cdkal1* KO podocytes.**

(A) Representative western blot images of CD2AP in *Cdkal1* KO or sgControl SVI podocyte cells following *Cd2ap* overexpression or mock treatment. (B) Albumin measurements in permeability assays using *Cd2ap*-overexpressed *Cdkal1* KO ($n = 8$), and mock-treated podocyte cells ($n = 16$). Data are presented as the mean ± SEM. *$P = 0.0388$ (KO1 Mock vs OE); **$P = 0.0073$ (sgControl Mock vs KO1 Mock) and **$P = 0.0017$ (sgControl Mock vs KO2 Mock) by two-way ANOVA followed by Dunnett's correction. (C) Representative images of F-actin-stained *Cdkal1* KO SVI cells overexpressing *Cd2ap* (in green). Scale bar, 20 μm. (D) The cellular area occupied by F-actin was quantified using *Cdkal1* KO SVI cells overexpressing *Cd2ap*. $n = 10$; data are presented as the mean ± SEM; *$P = 0.0395$ (KO1 Mock vs OE); ****$P = 9.71 \times 10^{-6}$ (KO2 Mock vs OE); **$P = 0.0011$ (sgControl Mock vs KO1 Mock) and **** $P = 3.27 \times 10^{-8}$ (sgControl Mock vs KO2 Mock) by two-way ANOVA, followed by Dunnett's correction. (E, F) Representative images of the results of the wound cell migration assay following mock treatment and *Cd2ap* overexpression in *Cdkal1* KO SVI podocyte cells at 0 h (upper panel) and after 30 h (lower panel). Scale bars, 500 μm. (G) Results of the wound cell migration assay using *Cd2ap*-overexpressed *Cdkal1* KO, and mock-treated podocyte cells. $n = 8$ each. Data are presented as the mean ± SEM; *$P = 0.0127$ (KO1 Mock vs OE); ****$P = 4.12 \times 10^{-7}$ (KO2 Mock vs OE); **$P = 0.0106$ (sgControl Mock vs KO1 Mock) and **** $P = 1.55 \times 10^{-9}$ (sgControl Mock vs KO2 Mock) by two-way ANOVA, followed by Dunnett's correction. Source data are available online for this figure.

insulin production or secretion between the risk allele group and the non-risk group of patients with DM (Fig. EV5C,D). This discrepancy may be explained by the participants in the clinical study having a relatively short duration of diabetes (a mean of 9 years). Overall, our preliminary human data suggest that *CDKAL1* variants may contribute directly to promote podocyte dysfunction and accelerate the progression of CKD, providing an important basis for a future large-scale human study.

The present study had several limitations. First, throughout most of the present study, we used mouse-derived materials, rather than human-derived materials, because obtaining the latter would require the biopsy of patient kidneys, which is highly invasive. In the future, we wish to conduct a clinical study to evaluate the causative links between *CDKAL1* mutations and podocyte dysfunction using genetic and pathologic approaches and human-derived materials. Second, further studies with a larger cohort and longer follow-up would be helpful to better understand the association between *CDKAL1* variants and kidney function.

In conclusion, we have demonstrated that a defect in tRNA modification reduces lysine translational efficiency, promotes podocyte dysfunction, and accelerates the progression of CKD. These findings present novel molecular mechanisms of CKD progression, paving the road towards reducing the numbers of patients waiting for renal replacement therapy.

# Methods

### Reagents and tools table

| Reagent/Resource | Reference or Source | Identifier or Catalog Number |
|---|---|---|
| **Experimental models** | | |
| E11 | Cytion | 400494 |
| SVI | Cytion | 400495 |
| **Recombinant DNA** | | |
| lentiCRISPR v2 BLAST | Addgene | 83480 |
| pSF-CMV-CMV-Sbf1-Fluc | Sigma-Aldrich | OGS608 |
| pLX302 | Addgene | 25896 |
| **Antibodies** | | |
| Rabbit anti-Cdkal1 polyclonal antibody | Proteintech | 22988-1-AP |
| Rabbit anti-podocin polyclonal antibody | Sigma-Aldrich | P0372 |
| Goat anti-αSMA polyclonal antibody | Gene Tex | 89701 |
| Rabbit anti-CD31 monoclonal antibody | Abcam | Ab222783 |
| Donkey anti-goat Alexa 488 | Invitrogen | A-11055 |
| Goat anti-rabbit Alexa 647 | Invitrogen | A-21244 |
| FlexAble Coralite Plus 647 antibody Labeling kit for Rabbit IgG | Proteintech | KFA003 |
| FlexAble Coralite Plus 488 antibody Labeling kit for Rabbit IgG | Proteintech | KFA001 |

| Reagent/Resource | Reference or Source | Identifier or Catalog Number |
|---|---|---|
| Phalloidin-iFluor 488 Reagent | Abcam | ab176753 |
| Rabbit anti-Cdkal1 ployclonal antibody | Invitrogen | PA5-29077 |
| Rabbit anti-Cd2ap ployclonal antibody | Gene Tex | GTX106235 |
| Rabbit anti-Synaptopodin polyclonal antibody | Proteintech | 21064-1-AP |
| Rabbit anti-ANLN antibody | Sigma-Aldrich | HPA050556 |
| Rabbit anti-NPHS2 antibody | Abcam | ab50339 |
| Rabbit anti-Gapdh polyclonal antibody | Gene Tex | GTX100118 |
| Goat anti-Rabit IgG Secondary antibody | Invitrogen | 31460 |
| Mouse anti-Ubiqutin | Cell Signal Technology | 3936S |
| **Oligonucleotides and other sequence-based reagents** | | |
| Mouse sgControl #1 S | 5'-CACCGGCGAGGTATTCGGCTCCGCG-3' | |
| #1 AS | 5'-AAACCGCGGAGCCGAATACCTCGCC-3' | |
| Mouse sgControl #2 S | 5'-CACCGGCTTTCACGGAGGTTCGACG-3' | |
| #2 AS | 5'-AAACCGTCGAACCTCCGTGAAAGCC-3' | |
| Mouse sgCdkal1 #1 S | 5'-CACCGACTTGATTGAGTTTCTAAAG-3' | |
| #1 AS | 5'-AAACCTTTAGAAACTCAATCAAGTC-3' | |
| Mouse sgCdkal1 #2 S | 5'-CACCGGCGAGGTATTCGGCTCCGCG-3' | |
| #2 AS | 5'-AAACCGCGGAGCCGAATACCTCGCC-3' | |
| Mouse sgCdkal1 F | 5'-TCCAATGCCTAGGACCTCAT-3' | |
| Mouse sgCdkal1 R | 5'-TAAGCAGGAACATCAGCGTG-3' | |
| Mouse *Cd2ap* F | 5'-AAGGAGAACTAAATGGGAGACGA-3' | |
| Mouse *Cd2ap* R | 5'-CCGTTTGATGGGCAAATTGTCA-3' | |
| **Chemicals, Enzymes and other reagents** | | |
| RPMI-1640 | Wako | 189-02025 |
| DMEM | Gibco | 1195-065 |
| PBS | Gibco | 10010-023 |
| Bovine serum albumin | Sigma-Aldrich | A7030-50G |
| Accutase | Nacalai Tesque | 12679-54 |
| Opti-MEM | Gibco | 31985-070 |
| Lipofecatmine 3000 | Thermo Fisher Scientific | L3000-015 |
| Fetal Bovine Serum | Capricorn Scientific | FBS-12A |
| TRI Reagent | Molecular Research Center | TR118 |
| Nuclease P1 | Wako | 145-08221 |

| Reagent/Resource | Reference or Source | Identifier or Catalog Number |
|---|---|---|
| TB Green Premix Ex TaqII | Takara | RR820L |
| PrimerScript RT Master Mix | Takara | RR036A |
| KOD FX | TOYOBO | KFX-101 |
| Alkaline Phosphatase C75 | Takara | 2120 A |
| DNase I | Roche | 04716728001 |
| RNase inhibitor | Nacalai Tesque | 30260-96 |
| T4 polynucleotide kinase | New England Biolabs | M0201L |
| Skim Milk Powder | Wako | 190-12865 |
| Glycine | Wako | 077-00735 |
| Tris | Wako | 207-06275 |
| Puromycin | Sigma-Aldrich | P8833-10MG |
| Blasticidin | Invivogen | BLL-42-02 |
| MG132 | Wako | 139-18451 |
| SYBR Gold | Invitrogen | S11496 |
| Gateway LR Clonase II Enzyme Mix | Invitrogen | 11791-020 |
| One Shot Stbl3 Chemically Competent Cells | Invitrogen | C7373-03 |
| Amersham ECL Prime Western Blotting Reagent | Cytiva | RPN2236 |
| Immunobilon-P transfer membrane | Millipore | IPVH00010 |
| O.C.T.Compund | Tissue–Tek | 4583 |
| Blocking solution | Nacalai Tasque | 06349-64 |
| Dual-Luciferase Reporter Assay System | Promega | E1910 |
| Streptozotocin | Wako | 191-15151 |
| Medetomidine hydrochloride | FUJIFILM | 139-17471 |
| Butorphanol | Meiji | 16 |
| Midazolam | Maruishi | 132324 |
| Antipamezole | ZENOAQ | 107050 |
| **Software** | | |
| GraphPad Prism 10.2 | https://www.graphpad.com/ | |
| JMP 9.0 | https://www.sas.com/en_us/home.html | |
| Image J | http://fiji.sc/ | |
| **Other** | | |
| LBIS Mouse Insulin ELISA Kit | Fujifilm | 634-01481 |
| QIAprep Spin Miniprep Kit | Qiagen | 27106 |
| QIAquick Gel Extraction Kit | Qiagen | 28706 |
| Qiagen Plasmid Max Kit | Qiagen | 12163 |
| QIAamp DNA Blood Mini Kit | Qiagen | 51104 |
| pENTR/D-TOPO Cloning Kit | Invitrogen | 45-0218 |
| BigDye Termination Sequencing Kit | Applied Biosystems | 4336917 |
| Pierce™ BCA Protein Assay Kit | Thermo scientific | 23225 |

| Reagent/Resource | Reference or Source | Identifier or Catalog Number |
|---|---|---|
| TaqMan SNP Genotype Assay Kit for rs7756992 | Life Technologies | 4351379 |
| TaqMan Genotyping Master Mix | Life Technologies | 4371353 |
| t$^6$A, $N^6$-threonylcarbamoyladenosine | SCB | sc-286478 |
| ms$^2$t$^6$A, 2-methylthio-$N^6$-threonylcarbanoyladenosine | TRC | M330590 |
| A, adenosine | WAKO | 015-24591 |
| HV-filter 0.45 μm | Millex | SLHV33RS |
| GV-filter 0.22 μm | Millex | SLGVR33RS |
| Stainless steel sieves 45 μm | SANPO | 5-3294-49 |
| Stainless steel sieves 106 μm | SANPO | 5-3294-42 |
| Stainless steel sieves 150 μm | SANPO | 5-3294-40 |
| Corning Biocoat Collagen I Cell Inserts | Corning | CLS354444 |
| Glass Base Dish | Iwaki | 3911-035 |
| ACCU-CHECK Avival Nano Blood Glucose Meter | Roche | 05075564002 |
| ACCU-CHECK Aviva Plus | Roche | 000317 |
| Metabolic cage | Tecniplast | 3600M021 |

## Transgenic mice, treatments, and 5/6 nephrectomy

The mice were housed at 25 °C under a 12-h light/dark cycle, with free access to food and water. Floxed mice were previously generated by the insertion of loxP sequences to flank exon 5 of the *Cdkal1* gene (Wei et al, 2011). Systemic *Cdkal1* KO mice were generated by mating the floxed mice with CAG-Cre mice provided by RIKEN, through the National Bioresource Project of the Ministry of Education, Culture, Sports, Science and Technology (MEXT) (Matsumura et al, 2004; Wei et al, 2011). To generate podocyte-specific *Cdkal1* KO mice, floxed mice were crossed with B6.Cg-Tg (*Nphs2*-Cre) 295Lbh/J mice purchased from the Jackson Laboratories (stock # 008205). Before being used for experiments, these mice were backcrossed for three generations. Serum samples were collected from a tail vein and urine samples were collected during housing in metabolic cages (Techniplast) for 24 h. Serum creatinine and urine biochemical parameters were measured by SRL Laboratories (Tokyo, Japan). Streptozotocin (STZ, Wako) was diluted to 40 mg/kg using saline and injected intraperitoneally into 8-week-old mice on 5 consecutive days. For glucose tolerance testing, mice were fasted for 7-hour (8:00 am to 3:00 pm), followed by an intraperitoneal injection of glucose (1 g/kg). Blood samples were obtained from the tail vein and glucose levels were determined using a glucometer (ACCU-CHEK, Roche) (Furman, 2021; Matsuzaka et al, 2004). Serum insulin levels were determined 30 min after the glucose challenge using an ELISA kit (LBIS Mouse Insulin ELISA Kit, Fujifilm). We confirmed that the blood glucose levels of the mice were >350 mg/dL (19.4 mmol/L) on two occasions. 5/6 nephrectomy was performed according to a previous report (Oikawa et al, 2010) in 8-week-old floxed mice, systemic

*Cdkal1* KO mice, and podocyte-specific *Cdkal1* KO mice. We performed right-sided nephrectomy, followed by left-sided 2/3 nephrectomy 2 weeks later. The procedures were performed under anesthesia with medetomidine hydrochloride (Fujifilm) 0.3 mg/kg, midazolam (Maruishi Pharmaceuticals) 4 mg/kg, butorphanol (Meiji Pharmaceuticals) 5 mg/kg, and atipamezole (Zenoaq) 0.75 mg/kg. To collect glomeruli, 20-week-old mice were sacrificed and their kidneys were collected. After mincing of the kidneys, we used stainless steel sieves (150, 106, and 45 μm) (SANPO) to collect the glomeruli, as previously described (Wang et al, 2019).

## Immunofluorescence

After sacrifice, the kidneys of mice were collected, snap-frozen in optimal cutting temperature compound (Tissue-Tek), and fixed in ice-cold acetone for 5 min. The sections were cut at 4 μm using a cryostat (Leica, CM1510S), washed three times with PBS (Gibco), and incubated in Blocking solution (Blocking One Histo, Nacalai Tesque) for 30 min at room temperature. For CDKAL1 single immunostaining, the sections were washed three times with PBS at room temperature, rabbit anti-CDKAL1 polyclonal antibody (1:75, Proteintech) was added, and the sections were incubated overnight at 4 °C. The following day, secondary goat anti-rabbit Alexa 647 (1:200, Invitrogen) antibody was added, and the sections were incubated for 1 h at RT in the dark. For double immunostaining, the sections were incubated overnight at 4 °C with a goat anti-αSMA polyclonal antibody (1:100, Gene Tex) and a rabbit anti-CDKAL1 polyclonal antibody (1:75, Proteintech). The next day, secondary donkey anti-goat Alexa 488 (1:200, Invitrogen) and goat anti-rabbit Alexa 647 (1:200, Invitrogen) antibodies were added, and the sections were incubated for 1 h at RT in the dark. For the concomitant staining for podocin/CDKAL1 and CD31/CDKAL1, FlexAble CoraLite Plus 488 (Proteintech, 1 μL) and FlexAble CoraLite Plus 647 (Proteintech, 3.5 μL) kits were used with rabbit anti-Podocin (0.5 μg, Sigma-Aldrich) and anti-CDKAL1 (1.75 μg, Proteintech) antibodies, respectively. Similarly, FlexAble CoraLite Plus 488 (Proteintech, 1 μL) and FlexAble CoraLite Plus 647 (Proteintech, 4.2 μL) kits were used with rabbit anti-CD31 (0.5 μg, Abcam) and anti-CDKAL1 (3 μg, Proteintech) antibodies, respectively, and the sections were incubated for 2 h at RT. The FlexAble CoraLite kits were used according to the manufacturer's instructions. Microscopic images were obtained using a confocal laser scanning microscope (Olympus, FV3000).

## Examination of the foot processes of podocytes by transmission electron microscopy

Under deep anesthesia induced by isoflurane inhalation, mice were fixed by perfusion through the ascending aorta using 2% paraformaldehyde (PFA) and 2.5% glutaraldehyde in 0.1 M phosphate buffer. Their kidneys were trimmed into 1-mm³ cubes. Post-fixation was performed with 1% $OsO_4$ in 0.1 M phosphate buffer on ice for 1 h. The tissue blocks were stained with 1.5% uranyl acetate, dehydrated through a graded ethanol series, and infiltrated with propylene oxide. Ultrathin sections (65-nm thick) were cut using an ultramicrotome (Leica, EM UC7) and stained with 1.5% uranium acetate and lead citrate. Images were captured using a transmission electron microscope (Hitachi, HT7700). The effacement of podocytes was assessed by quantifying the number of

foot processes per 100 μm of glomerular basement membrane using ImageJ (1.53v; NIH) (Martin et al, 2022), with five independent images analyzed from a single floxed mouse and a single KO mouse.

## Cell culture and KO cell generation

E11 and SVI murine kidney podocyte cell lines were purchased from Cytion (Eppelheim, Germany). The cells were cultured in RPMI medium (Wako) supplemented with 10% fetal bovine serum (FBS, Sigma-Aldrich) at 33 °C and induced to differentiate by culture at 37 °C for 2 weeks in a 5% $CO_2$ atmosphere (Hsu et al, 2017; Ito et al, 2017). We knocked out *Cdkal1* in the E11 and SVI cell lines using sense and antisense oligonucleotides encoding a single guide RNA (sgRNA) targeting exon 5 of *Cdkal1* that had been cloned into the BsmBI site of the Lenti-CRISPR v2 Blast vector (Addgene). Both cell lines were transduced with lentivirus made by transfecting HEK293FT cells with the sgRNA-containing Lenti-CRISPR v2 Blast, psPAX2 (Addgene), and pMD2.G (Addgene). Successfully transduced cells were selected by culture in medium containing 20 μg/mL blasticidin (Invivogen) for 3 days, then seeded into 10-cm dishes at 100 cells/dish. Single colonies were then selected using cloning disks (Sigma-Aldrich). sgControl cells were made by performing the same procedures using sgRNA oligo DNAs that do not target the mouse genome. The sgRNA-target region of the genome in each clone was PCR-amplified and sequenced. The loss of ms$^2$t$^6$A from the *Cdkal1* KO podocytes was confirmed by LCMS-8060NX system (Shimazu), according to a previously published method (Nagayoshi et al, 2022), as described below. *CDKAL1* KO HuH-7 cells were generated in the same way as E11 and SVI cells using appropriate sgRNA-encoding oligo DNAs. The oligonucleotide sequences are listed in Reagents and Tools Table.

## Luciferase reporter experiments

Cells were seeded on 96-well plates (7 × 10³ E11 cells/well) 1 day before transfection, and then transfected with 10 ng of codon reporter or control plasmids using Lipofectamine 3000 (Thermo Fisher Scientific). After 24 h, luciferase activity was measured using the Dual-Luciferase Reporter Assay System (Promega), a specialized 96-well plate (Greiner), and a Centro XS3 LB960 luminometer (Bethold Technologies). The translation reporter was constructed using a plasmid containing dual CMV promoters (pSF-CMV-CMV-Sbf1-Fluc, Sigma-Aldrich). cDNA encoding the *Renilla luciferase* gene was cloned into the XhoI and BamHI sites. Subsequently, oligonucleotides containing five consecutive AAA or AAG lysine codons, TTT phenylalanine codons, or five random codons for a control plasmid were ligated at the 5′-end of the *Renilla luciferase* gene using the NcoI and XhoI sites. Luciferase reporter experiments were performed essentially as described previously (Matsuura et al, 2024).

## Wound healing migration assay

Cells were seeded at 100,000 cells/well into a 12-well culture plate. After 12 h of growth, mechanical scraping was performed with a 200-μL pipette tip from the top to the bottom of the well, as previously described (Cechova et al, 2018). Images of each scratched area were obtained immediately after scraping (0 h) and after 30 h using a light

microscope (Olympus, CKX53). A magnification of 4× was used for imaging, and the scraped area at each time point was measured using ImageJ software (1.53v; NIH). The percentage of the area at 30 h relative to the area at 0 h for cells of each genotype was calculated. Five replicates were performed during this assay.

## Permeability assay

Ten thousand differentiated E11 cells were seeded onto type I collagen-coated 0.4 μm polycarbonate transwell filters in a 24-well filtration microplate (Corning). After 12 h, the culture medium was replaced with FBS-free RPMI medium in both the upper and lower chambers (Piwkowska et al, 2015). Following a 2 h-incubation, the medium in the lower chamber was replaced with RPMI medium supplemented with 40 mg/mL bovine serum albumin (BSA, Sigma-Aldrich), which had been filtered through a 0.22-μm filter (Millex), and the cells were incubated for an additional 2 h at 37 °C. The total protein concentration in the upper compartment was then quantified using a BCA protein assay kit (Thermo Fisher) (Cheng et al, 2012; Ishii et al, 2019).

## F-actin staining

Two hundred-fifty thousand E11 and SVI cells were seeded in 30-mm glass-based dishes. After 12 h, the cells were fixed in 4% PFA for 30 min at RT, then incubated with 0.1% Triton X-100 in PBS for 15 min. After two washes with PBS, the cells were incubated with Blocking solution (Nacalai Tesque) in PBS for 30 min, and then stained with Phalloidin-iFluor-488 (1:1000, Abcam) and 4′,6-diami-dino-2-phenylindole (DAPI) (1:1000, Dojindo) in the dark for 1 h at RT. After incubation, the dishes were washed three times with PBS, then microscopic images were obtained using the confocal laser scanning microscope (Olympus, FV3000). To quantify the cellular area occupied by F-actin in each cell, cellular regions of interest (ROI) were defined by manually outlining the cell contours in ImageJ. The F-actin channel was converted to a binary image using a fixed threshold range. For each ROI, the F-actin-positive area was measured using a binary image (Duarte et al, 2019; Yasuda et al, 2023).

## Western blotting

Cultured cells and sieved mouse glomeruli were pelleted by centrifugation and lysed in chilled lysis buffer (150 mM NaCl, 50 mM Tris-HCl pH 8.0, and 1% NP-40) containing protease inhibitor cocktail (Roche). The protein concentration of each lysate was determined using a BCA protein assay kit. Samples were then electrophoresed in 10% polyacrylamide-SDS gels and transferred to polyvinylidene fluoride membranes (Merck Millipore). The membranes were incubated with rabbit anti-CDKAL1 antibody (Invitrogen), rabbit anti-CD2AP antibody (Gene Tex) at 1:500 dilutions, rabbit anti-Synaptopodin antibody (Proteintech) at a 1:1000 dilution and rabbit anti-GAPDH antibody (Gene Tex) at a 1:20,000 dilution. The antibodies were diluted in 5% skim milk in TBST buffer (150 mM NaCl, 25 mM Tris-HCl pH 7.4, 2.7 mM KCl, and 0.05% Tween-20), added to the membranes, and incubated overnight at 4 °C. The membrane was washed with TBST, followed by incubation with the secondary antibody at room temperature for 1 h, and then rewashed with TBST. Signals were detected using Amersham ECL Prime Western Blotting Detection Reagent (GE

Healthcare) and an imager (ChemiDoc MP imaging system, Bio-Rad). To examine the accumulation of ubiquitinated proteins, SVI sgControl cells were treated with 10 μmol/L MG132 (Wako) for 6 h to inhibit proteasomal degradation.

## Quantitative proteomic analysis by liquid chromatography–tandem mass spectrometry

The cell lysates were digested by trypsin using the phase-transfer surfactant method, as previously described (Nagano et al, 2022). The digested samples were desalted and resuspended in 0.1% trifluoroacetic acid after drying. Digested peptide samples corresponding to 1 μg protein were subjected to LC–tandem MS using SWATH modes on a TripleTOF6600 (SCIEX) with the Eksigent NanoLC400 system (SCIEX). Proteins were identified and quantified using DIA-NN 1.8 and UniProt Human reference proteome data (Demichev et al, 2020). The quantified proteins are listed in Supplementary File. The raw data files generated during the proteomic analysis have been deposited in jPOST (http://jpostdb.org, jPOST ID: JPST003458/PXD057714). GO analysis was performed using genes corresponding to proteins showing at least four-fold higher or lower expression in *Cdkal1* KO E11 podocytes *vs*. Control cells in the Search Tool for the Retrieval of Interacting Genes/Proteins database, to obtain molecular functions and cellular components.

## Ribosome profiling

Ribosome profiling was essentially performed as described previously (Ingolia et al, 2009; Tresky et al, 2024) with minor modifications. Approximately $2.5 \times 10^6$ SVI podocytes in 10-cm dishes were washed with ice-cold PBS and harvested by scraping in 3 mL of lysis buffer [20 mM Tris-HCl (pH 7.5), 150 mM NaCl, 6 mM MgCl$_2$, 1 mM DTT, 1% Triton X-100, 150 μg/mL cycloheximide, 10 U/mL DNase I (Roche) and 1% RNase inhibitor (Nacalai)] on ice for 15 min. RNase I was added at a ratio of 1 μL per 3 μg of RNA (estimated by measuring the optical density (OD) using a Biospectrometer; Eppendorf) and incubated for 45 min on ice. This was followed by the addition of 10 μL of RNase inhibitor and centrifugation at 100,000 rpm for 2 h using a TLA110 rotor (Beckman Coulter) to pellet the ribosomes through a sucrose cushion (1 M sucrose, 20 mM Tris–HCl pH 7.5, 150 mM NaCl, 5 mM MgCl$_2$, 1 mM DTT and 100 μg/mL cycloheximide). RNA was extracted from the pellet using the TRI Reagent. Library preparation was performed in the same way as described previously (Tresky et al, 2024). Ribosome-protected mRNA fragments were used to generate sequencing libraries that were sequenced using MiSeq (Illumina). Using Galaxy (TheGalaxyCommunity, 2022), raw FASTQ files were adapter-trimmed, converted to FASTA format, and mRNA sequences were Bowtie2-selected. Three-nucleotide periodicity and P-site position of the 29 nt footprints were manually confirmed by blasting approximately 100 footprints. Within the 29 nt footprints, a 16–18 nt region corresponding to the A-site codon was extracted to compare ribosomal A-site occupancy between KO and sgControl samples.

## LC–MS analysis of t⁶A and ms²t⁶A

sgControl cells and *Cdkal1* KO cells were harvested and total RNA was isolated using TRI Reagent (Molecular Research Center), according to the manufacturer's protocol. The extracted RNA

(200 ng in 24 μL) was digested using a mixture of 2.5 U of nuclease P1 (Wako) in 5 mM ammonium acetate pH 5.3, 0.2 U alkaline phosphatase (Takara), and 20 mM HEPES–KOH pH 7.0. The samples were then incubated for 3 h at 37 °C and subjected to mass spectrometric analysis (Ahmad et al, 2024). $t^6A$, $ms^2t^6A$, and unmodified adenosine were measured, and adenosine in 2-μL samples were measured using the same LC–MS, as previously described (Nagayoshi et al, 2022). The detection of modified nucleosides was confirmed using chemically synthesized $t^6A$ and $ms^2t^6A$, the details of which are shown in Reagents and Tools Table.

### *Cd2ap* overexpression experiments using lentivirus

Untagged mouse *Cd2ap* cDNA was cloned into the pDONR221 vector (Thermo Fisher). The cDNA was transferred into the pLX302 vector (Addgene) using the Gateway LR Clonase II enzyme mix (Invitrogen), according to the standard protocol. The construct was transformed into and expanded using One Shot Stbl3 chemically competent *Escherichia coli* (Thermo Fisher). The amplified recombinant lentiviral DNA was transfected into HEK293FT cells with psPAX2 (Addgene) and pMD2.G (Addgene) using Lipofectamine 3000 reagent. The medium was collected after 2 days, then passed through a 0.45-μm filter (Millex). The viruses were added to E11 or SVI cells, and colony selection was performed with puromycin (1.5 μg/mL) (Sigma-Aldrich) over 2 days.

### Statistics

In mouse and cell experiments, Student's *t* test and the Mann–Whitney *U* test were used to compare two groups, and multiple groups were compared using one-way or two-way ANOVA if the data were normally distributed, and the Kruskal–Wallis test, followed by Dunn's multiple comparison test, if not. For pairwise comparisons, post-hoc testing was performed using Tukey's correction. Normally distributed data with homogenous variance are expressed as the mean ± standard error of mean (SEM). The data were analyzed using GraphPad Prism 10.2 software. $P < 0.05$ was considered to indicate statistical significance.

### Study approval

The animal experiments and procedures were performed in accordance with the Guidelines for the Care and Use of Laboratory Animals, and were reviewed and approved by the Animal Ethics Committee of Kumamoto University (approval ID A21-103). This study adheres to the ARRIVE (Animal Research: Reporting of In Vivo Experiments) guidelines to promote transparent and comprehensive reporting of animal research.

## Data availability

All the data and cell resources presented in this study will be made available upon reasonable request to the corresponding author and the institutional review board of Kumamoto University.

The source data of this paper are collected in the following database record: biostudies:S-SCDT-10_1038-S44318-026-00759-3.

## Peer review information

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

## Acknowledgements

We thank the members of the Tomizawa laboratory for fruitful discussions. We are grateful to Yuka Tashiro, Nobuko Maeda, Mako Hirano and Akiko Chijiiwa for technical assistance, and Mark Cleasby, PhD, of BioEdit and Asma Begum, MSc, of Editage for their critical reading and language editing. A synopsis was created using BioRender (http://BioRender.com/k6o3rax). This work was supported by the Japan Society for the Promotion of Science KAKENHI grants 21K20925, 22K16224 (awarded to YN), and 25K02702 (awarded to KT), a Research Support Project for Life Science and Drug Discovery (Basic for Supporting Innovative Drug Discovery and Life Science Discovery (BINDS)) from AMED (grant number JP23ama121018; awarded to SO), the Kumamoto University Center for the Metabolic Regulation of Health Aging (CMHA), and the Kumamoto University Diversity Promotion Office.

## Author contributions

**Hiroko Nagata**: Conceptualization; Investigation; Writing—original draft. **Yu Nagayoshi**: Conceptualization; Project administration. **Takeshi Chujo**: Conceptualization; Supervision; Writing—original draft; Writing—review and editing. **Hitomi Kaneko**: Supervision; Investigation. **Kayo Nishiguchi**: Investigation. **Yutaka Kakizoe**: Investigation. **Hiroko Ijima**: Resources; Investigation. **Korin Sakakida**: Resources; Investigation. **Takeshi Masuda**: Investigation; Methodology. **Sumio Ohtsuki**: Investigation; Methodology; Project administration. **Fan-Yan Wei**: Resources; Investigation; Methodology. **Yukie Takahashi**: Investigation. **Takaichi Fukuda**: Investigation. **Hideaki Jinnouchi**: Resources. **Yuki Adachi**: Investigation. **Ryosuke Yamamura**: Investigation. **Koki Matsushita**: Investigation. **Masataka Adachi**: Investigation. **Hideki Yokoi**: Investigation. **Kimitoshi Nakamura**: Supervision. **Hitoshi Nakazato**: Supervision; Investigation. **Kazuhito Tomizawa**: Conceptualization; Supervision; Writing—review and editing.

Source data underlying figure panels in this paper may have individual authorship assigned. Where available, figure panel/source data authorship is listed in the following database record: biostudies:S-SCDT-10_1038-S44318-026-00759-3.

## Disclosure and competing interests statement

The authors declare no competing interests.

# Expanded View Figures

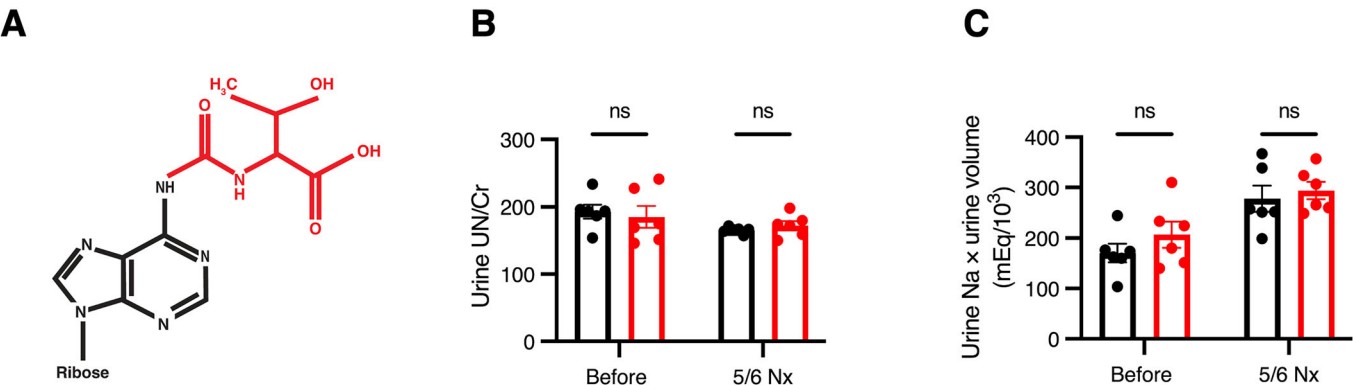

**Figure EV1. Lack of tubular injury after 5/6 nephrectomy in systemic *Cdkal1* KO mice.**

(**A**) Chemical structure of $N^6$-threonylcarbamoyladenosine ($t^6A$). Modified residue is shown in red. (**B**, **C**) Urinary urea nitrogen (UN) level normalized by urine creatinine (Cr) (**B**) and daily sodium excretion (**C**) in systemic *Cdkal1* KO and floxed mice before (8-week-old) and after 5/6 nephrectomy (12-week-old). $n = 6$ each. Data are presented as the mean ± SEM. n.s., not significant by two-way ANOVA followed by Sidak post-hoc test.

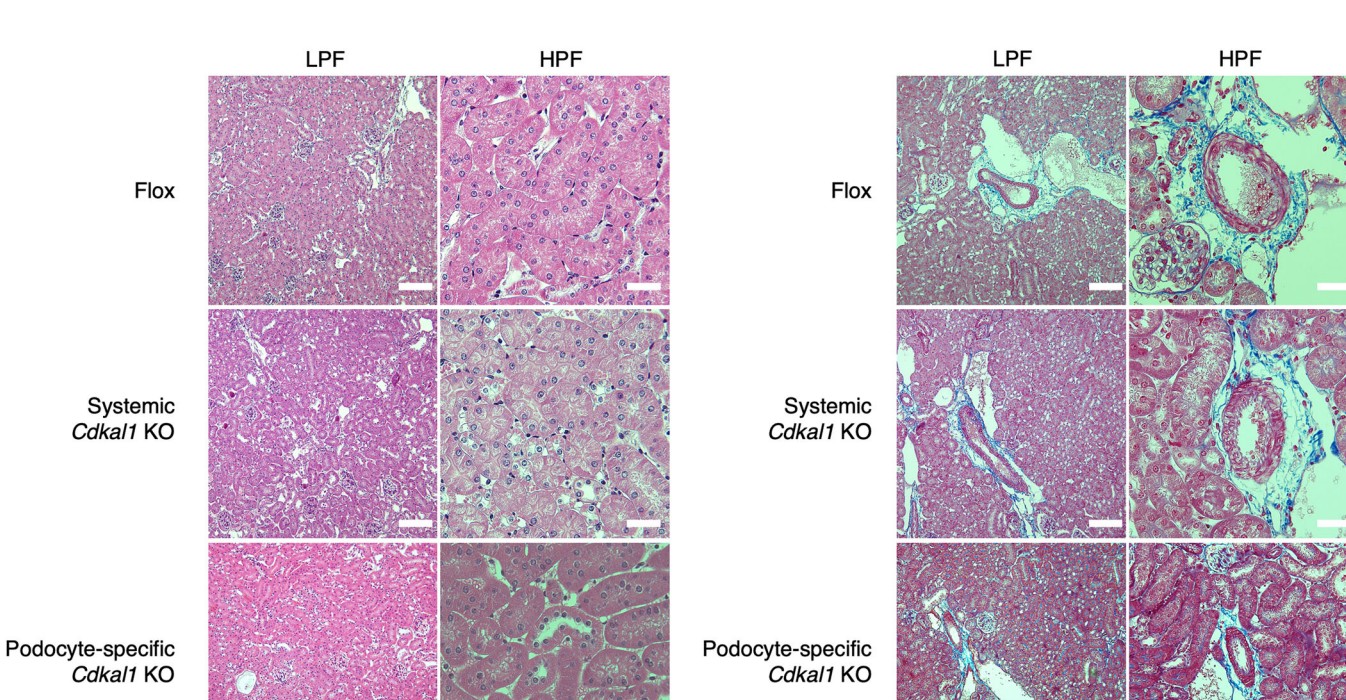

**Figure EV2.   Morphology of the renal tubules and vessels of the floxed mice, systemic *Cdkal1* KO mice, and podocyte-specific *Cdkal1* KO mice.**

(A) HE stained renal tubules. (B) Azan–Mallory stained renal vessels of systemic *Cdkal1* KO mice, podocyte-specific *Cdkal1* KO, and floxed mice. Scale bars, 200 μm (low-power fields, LPF) and 50 μm (high-power fields, HPF).

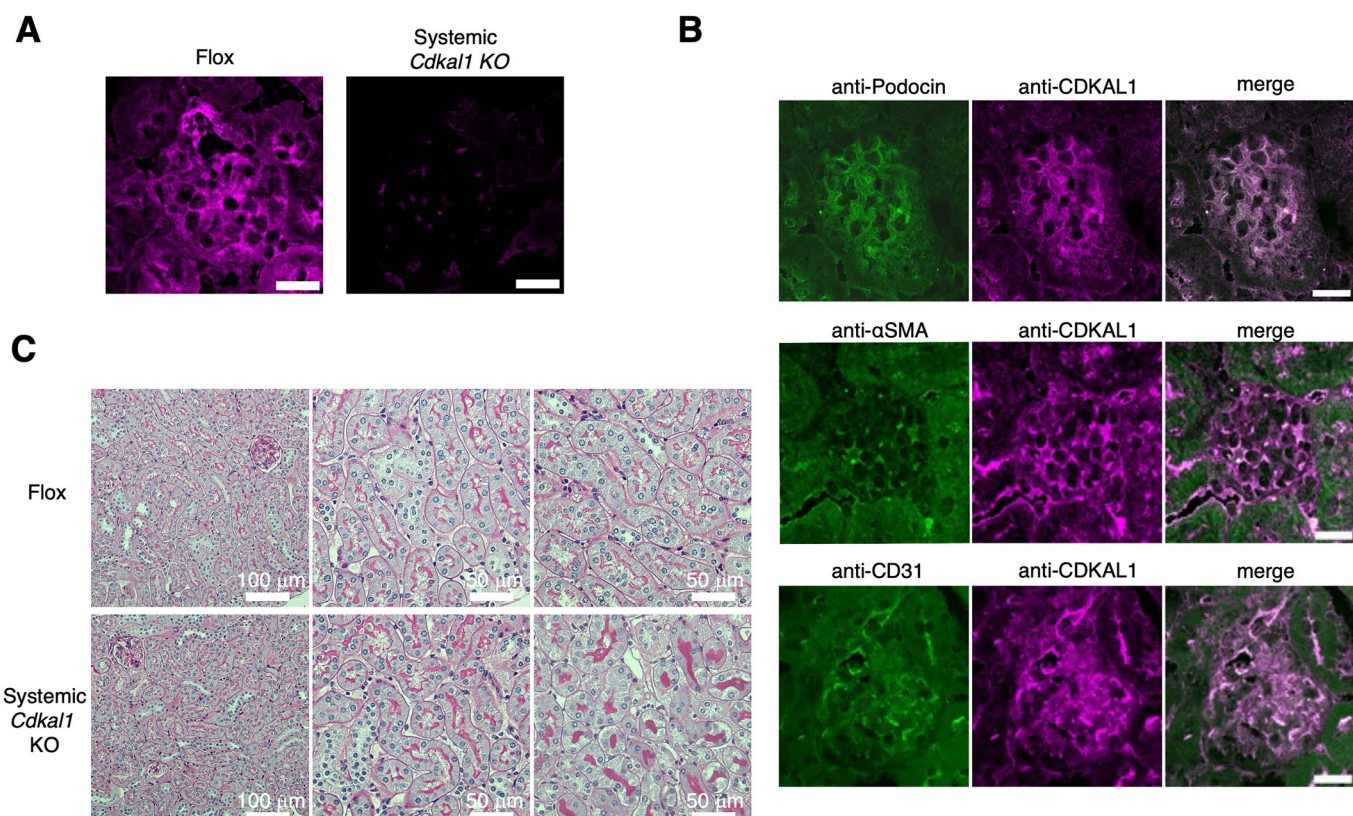

**Figure EV3.   Validation, localization and renal phenotype of CDKAL1 in mouse kidneys.**

(**A**) Immunostaining with anti-CDKAL1 antibody shows no signal in glomeruli from systemic *Cdkal1* knockout (KO) mice, confirming the antibody's specificity. Scar bar, 20 μm. (**B**) Double immunofluorescence staining for podocin, αSMA or CD31 (green) with CDKAL1 (magenta) of the glomeruli of 20-week-old male mice. αSMA and CD31 are markers of mesangial cells and endothelial cells, respectively. Scale bars, 20 μm. (**C**) Representative PAS-stained renal tubules of 50-week-old systemic *Cdkal1* KO or floxed mice. Scale bars, 100 μm (left), 50 μm (center, right).

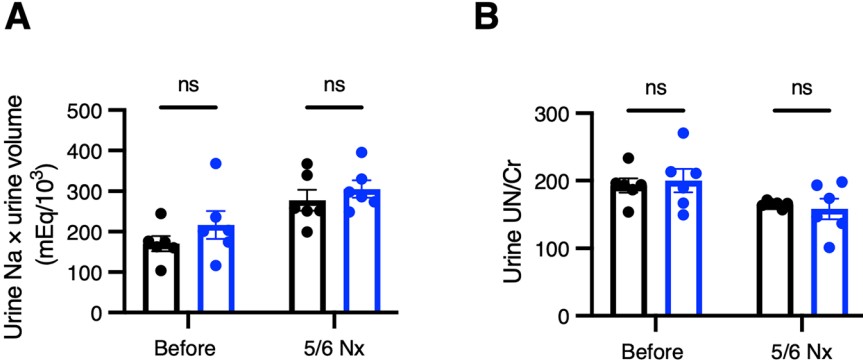

**Figure EV4. No tubular injury in podocyte-specific *Cdkal1* knockout mice after 5/6 nephrectomy.**

(A, B) Measurement of urinary urea nitrogen (UN) normalized by urine creatinine (Cr) (A) and daily sodium excretion (B) in podocyte-specific *Cdkal1* KO and floxed mice before (8-week-old) and after 5/6 nephrectomy (16-week-old). *n* = 6 each. Data are presented as mean ± SEM; n.s., not significant by two-way ANOVA followed by Sidak post-hoc test.

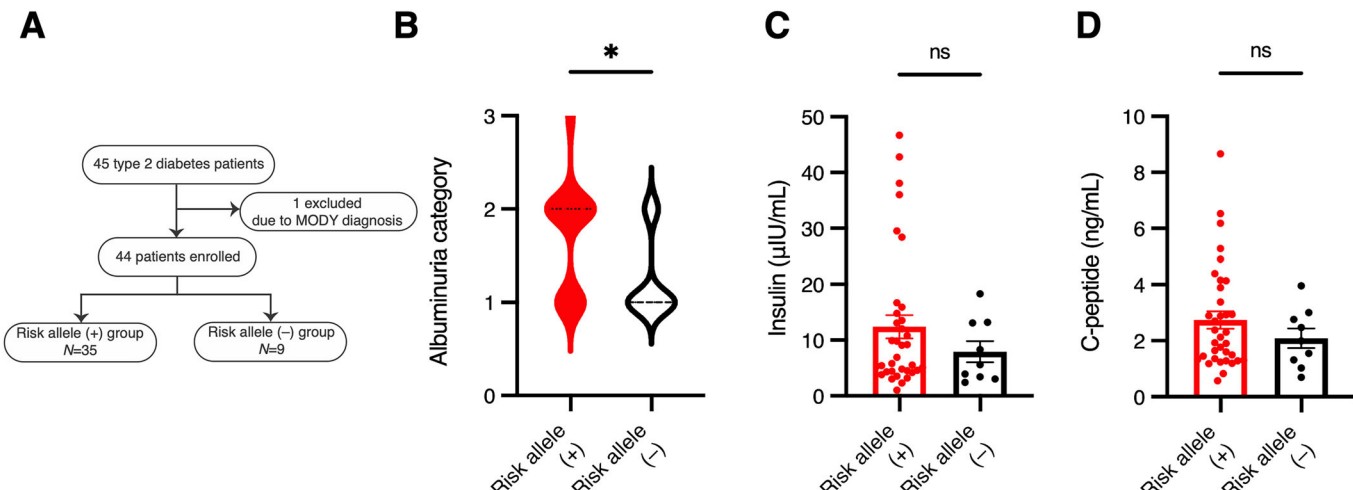

**Figure EV5. Earlier onset of albuminuria in patients with type 2 DM and a *CDKAL1* SNP than in those without.**

(**A**) Enrollment of patients in the clinical study. Out of 45 initial participants, one was excluded, resulting in 44 patients being analyzed. MODY, maturity onset diabetes of the young. (**B–D**) Albuminuria categories (**B**), plasma insulin levels (**C**), serum C-peptide levels (**D**) in patients with ($n = 35$) or without ($n = 9$) the risk allele. Albuminuria categories according to the KDIGO 2022 Clinical Practice Guideline (KDIGO, 2022). Data are presented as mean ± SEM. Statistical significance was determined by Mann–Whitney $U$ test: *$P = 0.0264$ (**B**); n.s., not significant (**C**, **D**).

