## [Peer Review File · The EMBO Journal]

CDKAL1 dysfunction impairs lysine codon translation in podocytes and accelerates chronic kidney disease

Hiroko Nagata, Yu Nagayoshi, Takeshi Chujo, Hitomi Kaneko, Kayo Nishiguchi, Yutaka Kakizoe, Hiroko Ijima, Kori Sakakida, Takeshi Masuda, Sumio Ohtsuki, Fan-Yan Wei, Yukie Takahashi, Takaichi Fukuda, Hideaki Jinnouchi, Yuki Adachi, Ryosuke Yamamura, Koki Matsushita, Masataka Adachi, Hideki Yokoi, Kimitoshi Nakamura, Hitoshi Nakazato, and Kazuhito Tomizawa

Corresponding author(s): Kazuhito Tomizawa (tomikt@kumamoto-u.ac.jp)

Review Timeline:

Submission Date:	26th May 25
Editorial Decision:	30th Jul 25
Revision Received:	27th Nov 25
Editorial Decision:	30th Jan 26
Revision Received:	7th Feb 26
Accepted:	5th Mar 26

Editor: Cornelius Schneider

Transaction Report:

Dear Prof. Tomizawa,

Thank you for submitting your manuscript for consideration by the EMBO Journal. It has now been seen by two referees whose comments are shown below. We were not able to obtain the feedback of the third referee until now and we have therefore decided to not wait any further and proceed with these two reports. We will share the missing report with you in case we still receive the feedback but will only take it into consideration for our assessment if it voices critical technical concerns.

As you can see from the reports both referees are interested in the topic and think that the overall quality of the manuscript is high. However, referee #1 voices several major concerns requiring additional controls and proposes experiments which in my view would significantly strengthen the manuscript. I would therefore like to invite you to submit a revised version of the manuscript, addressing the comments of both reviewers including the experiments proposed in points 2 and 4 of referee #1. I should add that it is EMBO Journal policy to allow only a single round of revision, and acceptance of your manuscript will therefore depend on the completeness of your responses in this revised version.

Thank you for the opportunity to consider your work for publication. Please do not hesitate to contact me if you have any additional questions. I am also happy to discuss the revisions and potential experiments by e-mail or video conferencing. I look forward to your revision.

Yours sincerely,

Cornelius Schneider, PhD
Editor
The EMBO Journal
c.schneider@embojournal.org

Please see our instructions to authors

- a Reagents and Tools Table as part of the Methods section, which can be downloaded from our author guidelines

We realize that it is difficult to revise to a specific deadline. In the interest of protecting the conceptual advance provided by the work, we recommend a revision within 3 months (28th Oct 2025). Please discuss the revision progress ahead of this time with the editor if you require more time to complete the revisions. Use the link below to submit your revision:

Referee #1:

In their manuscript "CDKAL1 dysfunction impairs lysine translation in podocytes and accelerates chronic kidney disease" Nagata and co-authors, using systemic and podocyte-specific *Cdkal1* knockout mice and cells, demonstrate that chronic kidney disease may result from the absence of m^2t^6A at position 37 of tRNA^{Lys}UUU, which impairs lysine codon translation and consequently alters the renal proteome.

The manuscript is well-written and logically structured. While the experimental models are coherently designed, some important controls are missing. Moreover, direct evidence for translation defects specifically at lysine codons is lacking. The authors should consider performing ribosome profiling on the affected kidney tissue or, at the very least, in the KO cell culture model.

I could not find the list of references. The authors haven't included it in the pdf.

The following major points should be addressed to consider publication in The EMBO Journal:

- 1) Figure 2: Podocyte-specific *Cdkal1* KO mice: The specific ablation of *Cdkal1* in podocytes is not demonstrated. Although *Cre-Nphs2* is reported to be podocyte-specific, the authors should include controls confirming that pancreatic function and insulin production are unaffected in these mice and that the impairment of kidney function is not due to dysfunction in other organs.
- 2) The abnormality in F-actin polymerization and cellular migration defects are interesting in vitro observations. However, these are not confirmed in vivo in the authors' mouse models, and the underlying cause of these defects remains unexplained. Since actin is ubiquitously expressed, it is unclear why abnormalities would be restricted to podocytes. Knocking out *Cdkal1* in non-kidney cell lines could help determine whether the observed effect is direct or indirect. Additionally, the qualitative data presented in Figures 3i and 5c requires quantification.
- 3) Figure 4D, E, F: The authors should show the percentage of all other codons in the analysed protein datasets and discuss this in relation to m^2t^6A modification.
- 4) The effect observed in the proteome at lysine codons may be indirect, potentially caused by ubiquitination (on lysine residues) of misfolded or mistranslated proteins and the subsequent degradation of lysine rich proteins. Ribosome profiling would determine whether there is a direct impact on lysine codons during translation.
- 5) Panels 4I and K need quantification

Minor:

Panels 1E, F, G and 2C, D, F and 3C are missing definitions of the color codes used (black, red, blue).

Referee #2:

Nagata et al. characterized the function of CDKAL1, a tRNA-modifying enzyme that is involved in the tRNA Lys biogenesis in renal glomerular podocytes. As a SNP of this gene is known to be associated with and contributes to the T2DM and CKD progression, they showed that podocyte-specific deletion of CDKAL1 resulted in a higher level of proteinuria and renal

glomerular injury in 5/6 nephrectomy and STZ-induced T1D models of mice. This clearly demonstrated an essential role of CDKAL1 deficiency in podocytopathy in CKD and DKD. They further performed LC-MS proteomic analyses and showed that a group of proteins essential for podocyte function have been affected by the lack of CDKAL1, presumably due to the lack of tRNA-Lys and hence reduced protein translational efficiency. Out of these proteins, CD2AP is mostly affected and overexpression of CD2AP could rescue partially the podocyte defects caused by loss of CDKAL1 in vitro. The data in this manuscript are clearly convincing and support the conclusion. Some clarification of methods and data quantification would help strengthen this manuscript. Although there is a lack of relevant data from human samples to support the finding, the authors have acknowledged this drawback. Overall, this manuscript reported another example of tRNA modopathy in the context of podocytopathy and CKD, so it would be of general interest for nephrology and kidney research fields. Given it is specifically focusing on renal diseases, such a report may be of a medium level of interest to the general readership of The EMBO journal.

Major Concerns: NONE

Minor concerns:

- 1) In Figure 1j-k, the authors showed foot process effacement was observed in aged *Cdkal1* mice. However, it is unclear whether the quantitative data were collected from 5 mice or 5 glomeruli of a single animal in each group. This must be clarified in the methods section. I also recommend showing the proteinuria assay results of the aged mice. renal tubular pathology is recommended for systemic KO mice as well.
- 2) In Figure 2, I would recommend examining a few IF or IHC of podocyte specific proteins to demonstrate the molecular pathology of renal glomeruli in podocyte-specific KO mice.
- 3) In Figure 3 and 5, F-actin staining needs to be quantified. Showing only a few representative images is sufficient.
- 4) In Figure 4, the authors only examined and confirmed CD2AP expression was reduced by KO of CDKAL1, while in Figure 4g, a list of proteins are rich in Lysine. Can the authors examine a few more of these proteins to strengthen the conclusion?

Responses to reviewers

Manuscript ID: EMBOJ-2025-121381.

Title: CDKAL1 dysfunction impairs lysine translation in podocytes and accelerates chronic kidney disease

To the Editor:

Thank you for handling our manuscript and providing us with an opportunity for revision. Your comments helped us improve the manuscript. We have carefully read the comments from you and the reviewers and revised the manuscript accordingly.

Referee #1:

1) **Figure 2: Podocyte-specific *Cdkal1* KO mice: The specific ablation of *Cdkal1* in podocytes is not demonstrated. Although *Cre-Nphs2* is reported to be podocyte-specific, the authors should include controls confirming that pancreatic function and insulin production are unaffected in these mice and that the impairment of kidney function is not due to dysfunction in other organs.**

We thank the reviewer for pointing this out. In response to the reviewer's comment, we performed a glucose tolerance test and insulin measurement in podocyte-specific *Cdkal1* KO mice and confirmed that glucose tolerance and insulin secretion were normal, as shown below. The data have been added as new Fig. 2B and C, and the relevant text has been added to the Results and Methods sections (page 9, lines 183–191; page 19, lines 443–448).

Figure 2. CKD progression phenotype observed upon increased kidney load in podocyte-specific *Cdkal1* KO mice. (B, C) Blood glucose (B) and serum insulin levels (C) during glucose tolerance test at 20 weeks. $n = 6$ for glucose and $n = 4$ for insulin. Data are presented as mean \pm SEM; n.s., not significant by Mann–Whitney U test.

2) **The abnormality in F-actin polymerization and cellular migration defects are interesting in vitro**

observations. However, these are not confirmed *in vivo* in the authors' mouse models, and the underlying cause of these defects remains unexplained. Since actin is ubiquitously expressed, it is unclear why abnormalities would be restricted to podocytes. Knocking out *Cdkal1* in non-kidney cell lines could help determine whether the observed effect is direct or indirect. Additionally, the qualitative data presented in Figures 3i and 5c requires quantification.

We appreciate the referee's comment. To address this, we examined the effect of *Cdkal1* KO on F-actin polymerization in HuH-7 cells, a human hepatocellular carcinoma cell line (Nakabayashi *et al*, 1982). *CDKAL1* KO HuH-7 cells were generated using the CRISPR–Cas9 system, and the loss of ms^2t^6A , the CDKAL1-dependent tRNA modification, was confirmed in HuH-7 *CDKAL1* KO cells using LC–MS analysis (Appendix Fig. S5A). Notably, *CDKAL1* KO HuH-7 cells showed no difference in F-actin localization patterns compared to sgControl cells (Appendix Fig. S5B, S5C). Based on these findings and given that Cd2ap is preferentially expressed in podocytes and colocalizes with F-actin (Lehtonen *et al*, 2002; Tang & Briehner, 2013), we speculated that the reduction of lysine-rich Cd2ap might contribute to the decreased F-actin levels observed in *Cdkal1* KO podocytes.

Second, the F-actin images shown in Figs. 3I and 5C were quantified, and the quantification results were added to the new Figs. 3J, K, and 5D, as shown on the next page. We have also added the quantification method to the Methods section (page 23, lines 551–555).

Appendix Figure S5. *CDKAL1* KO has no effect on F-actin organization in HuH-7 hepatocellular carcinoma cells. (A) Loss of ms^2t^6A modification within the total RNA of *CDKAL1* KO HuH-7 cells, as confirmed by LC–MS. (B) Representative images of F-actin stained with phalloidin in *CDKAL1* KO HuH-7 cells. Scale bar, 20 μ m. (C) Quantification of the cellular area

occupied by F-actin. $n = 10$; data are presented as the mean \pm SEM; n.s., not significant by one-way ANOVA, followed by Dunnett's correction.

Figure 3. *Cdkal1* KO leads to impaired lysine codon translation, filtration barrier dysfunction, and podocyte motility defect. (J, K) Quantification of the cellular area occupied by F-actin. J: E11, K: SVI. $n = 10$; data are presented as the mean \pm SEM; **** $P < 0.0001$ by one-way ANOVA, followed by Dunnett's correction.

Figure 5. *Cd2ap* overexpression partially restores filtration barrier function and migration in *Cdkal1* KO podocytes. (D) The cellular area occupied by F-actin was quantified *Cdkal1* KO SVI cells overexpressing *Cd2ap*. $n = 10$; data are presented as the mean \pm SEM; * $P = 0.0395$ (KO1 Mock vs OE); **** $P < 0.0001$ (KO2 Mock vs OE); ** $P = 0.0011$ (sgControl Mock vs KO1 Mock) and **** $P < 0.0001$ (sgControl Mock vs KO2 Mock) by 2-way ANOVA, followed by Dunnett's correction.

3) Figure 4D, E, F: The authors should show the percentage of all other codons in the analyzed protein datasets and discuss this in relation to m^2t^6A modification.

Thank you for this comment. We calculated the codon usage frequencies for the target proteins from the proteome analysis and assessed correlation of protein expressional changes upon *Cdkal1* KO to codon-usage frequencies of the proteins (Appendix Fig. S8). The analysis revealed that protein decreased in *Cdkal1* KO cells containing higher ratios of Lys AAA/AAG codons (Fig. 4D–F), as well as higher ratios of amino acids encoded by near-cognate codons to Lys AAA/AAG, such as Glu GAA/GAG. The percentage of all codons in the protein set identified in the proteomic dataset are shown in Appendix Fig. S9 (next page). The codons were arranged in the order of the codon table. Light blue dots indicate proteins that were decreased in *Cdkal1* KO podocytes compared to sgControl podocytes, whereas pink dots indicate proteins that were increased. Lys_AAA (Fig. 4E) and Lys_AAG (Fig. 4F) are indicated by the red squares. Codons that exhibited the same pattern as Lys_AAA and Lys_AAG are highlighted in yellow, whereas those showing the opposite pattern are highlighted in blue. These results might be due to the loss of ms^2 modification at position 37 of $tRNA^{Lys}_{UUU}$; $tRNA^{Lys}_{UUU}$ might show reduced codon recognition and more promiscuously bind to some of the near-cognate codons other than the cognate Lys AAA/AAG codons. However, some codons cannot be explained by this simple hypothesis, including Asp GAU/GAC (% of which decreased in increased proteins upon *Cdkal1* KO) and Pro CCA/CCU/CCG/CCC (% of which decreased in increased proteins upon *Cdkal1* KO). Thus, we might have to consider additional possibilities, such as ms^2 -less $tRNA^{Lys}_{UUU}$ promiscuously recognizing other codons or ribosome traffic jams that affect different codons. However, investigating such hypotheses would require extensive experiments such as quantification of tRNAs contained in translating ribosomes, di-ribosome profiling, and cryo-EM structural analysis of ribosome–tRNA–mRNA ternary complex, and may be a good subject for a future study.

We have added the amino acid composition analysis data to the new Appendix Fig. S8 and S9, and the corresponding discussion has been added to the Discussion section (page 15, lines 340–351).

Appendix Figure S8. Pearson correlation between codon-usage frequency and protein changes among upregulated or downregulated proteins in *Cdkal1* KO cells. Lysine codons are indicated

in red and their near-cognate codons are indicated in yellow. **** $P < 0.0001$ (Lys_AAG), **** $P < 0.0001$ (Asp_GAU), *** $P = 0.0001$ (Lys_AAA), *** $P = 0.0002$ (Glu_GAA), *** $P = 0.0066$ (Ile_AUU), * $P = 0.0106$ (Met_AUG), * $P = 0.0263$ (Glu_GAG), * $P = 0.0265$ (Thr_ACC), * $P = 0.0203$ (His_CAC), * $P = 0.0103$ (Gly_GGA), ** $P = 0.0049$ (Trp_UGG), ** $P = 0.0037$ (Cys_UGU), ** $P = 0.0012$ (Gly_GGG), *** $P = 0.0008$ (Pro_CCC), *** $P = 0.0004$ (Gly_GGC), *** $P = 0.0003$ (Pro_CCU), **** $P < 0.0001$ (Cys_UGC), and **** $P < 0.0001$ (Pro_CCA) by Pearson correlation test. No significant changes were observed in unindicated codons.

(This figure continues on the following page.)

Appendix Figure S9. Relative codon composition of proteins downregulated and upregulated in *Cdkal1* KO cells.

4) The effect observed in the proteome at lysine codons may be indirect, potentially caused by ubiquitination (on lysine residues) of misfolded or mistranslated proteins and the subsequent degradation of lysine rich proteins. Ribosome profiling would determine whether there is a direct impact on lysine codons during translation.

We appreciate the reviewer's insightful feedback on this aspect. Western blot analysis of ubiquitin revealed no significant differences in global ubiquitination levels between *Cdkal1* KO and control podocytes (Appendix Fig. S10). However, this result does not preclude the possibility of some lysine-rich proteins from being targeted to ubiquitin-mediated degradation. We recognize that investigation of such possibility is important, but we also believe that such analysis is beyond the scope of this study that primarily focuses on protein synthesis, at least within the limited revision period. The corresponding explanation has been added to the Discussion section (page 15, lines 353–358). We also performed ribosome profiling in *Cdkal1* KO, expecting to observe increased A-site occupancy at Lys AAA/AAG codons; however, no such increase was detected. Nevertheless, lysine codon reporter experiments clearly demonstrated that lysine translation was decreased in *Cdkal1* KO (Fig. 3C). One possible explanation for this discrepancy is that tRNAs that decode near-cognate codons may be misincorporated into the ribosomal A-site for lysine AAA/AAG codons. However, investigating this possibility requires extensive experiments, such as the quantification of tRNAs contained in translating ribosomes, and may be a good subject for future studies. The ribosome profiling result and the discussions were added to the Appendix Fig. S11 and Discussion section (page 15, lines 358–366), disclosing the limitations of our study and wishing that the result may provide clues to future studies.

Appendix Figure S10. Ubiquitination in *Cdkal1* KO podocytes. Western blotting analysis of ubiquitinated proteins in *Cdkal1* KO and control podocytes. SVI sgControl were treated with MG132 as a positive marker. GAPDH was used as a loading control.

Appendix Figure S11. Ribosome A-site occupancy across mRNA codons in WT and *Cdkal1* KO SVI podocytes (mean \pm SEM, $n = 4$ per group). Lysine codons are indicated in red and their near-cognate codons are indicated in yellow. $*P = 0.02249$ (Thr_ACG), $*P = 0.0297$ (Phe_TTT) by Welch's t -test. No significant changes were observed in unindicated codons.

5) Panels 4I and K need quantification.

We appreciate the reviewer's comments. In response to the reviewer's comment, we quantified the relative CD2AP protein levels normalized to GAPDH and incorporated the results into the new Fig. 4I, L, and N.

Figure 4. Higher lysine ratio in proteins that decreased in *Cdkal1* KO podocyte cell lines, and reduction of lysine-rich protein CD2AP in *Cdkal1* KO podocytes.

(I, L) Quantification of CD2AP protein levels normalized to GAPDH in *Cdkal1* KO and control E11 (I) and SVI (L) podocyte cells. $n = 4-5$ each. Data are presented as the mean \pm SEM. $**P = 0.0040$ (E11 sgControl vs KO1); $**P = 0.0017$ (E11 sgControl vs KO2); $*P = 0.029$ (SVI sgControl vs KO1); $**P = 0.0015$ (SVI sgControl vs KO2) by one-way ANOVA, followed by Dunnett's correction. (N) Quantification of CD2AP protein levels normalized to GAPDH in M. $n = 11$. Data are presented as the mean \pm SEM; $*P = 0.040$ by Mann–Whitney U test.

Minor:

Panels 1E, F, G and 2C, D, F and 3C are missing definitions of the color codes used (black, red, blue).

We thank the reviewer for pointing this out. We apologize for this omission and have added appropriate color coding to the figure.

Referee #2:

Major Concerns: NONE

Minor concerns:

1) In Figure 1j-k, the authors showed foot process effacement was observed in aged *Cdkal1* mice. However, it is unclear whether the quantitative data were collected from 5 mice or 5 glomeruli of a single animal in each group. This must be clarified in the methods section. I also recommend showing the proteinuria assay results of the aged mice. renal tubular pathology is recommended for systemic KO mice as well.

Thank you for pointing this out. For Fig. 1J and K, quantification was performed on five glomeruli obtained from one floxed mouse and one KO mouse. We have clarified this point in the Figure legends and Methods sections (page 21, lines 489–491; page 29, line 693).

The effacement of podocytes was assessed by quantifying the number of foot processes per 100 μm of glomerular basement membrane using ImageJ (1.53v; NIH), with five independent glomeruli analyzed from a single floxed mouse and a single KO mouse (page 21, lines 489–491).

Second, we appreciate the reviewer's recommendation and have now included the albuminuria results of aged mice and kidney tubular pathology in Fig. 1L and Expanded View Fig. 3C. A mild increase in albuminuria was detected in systemic *Cdkal1* KO mice at 40 weeks (Fig. 1L), whereas PAS staining revealed no tubular abnormalities (EV3C). These results and the irregular morphology of foot processes (Fig. 1J) collectively demonstrate that *Cdkal1* KO drives kidney

dysfunction in later adulthood by compromising podocytes and not tubules. Relevant descriptions were added to the Results section (page 8, lines 168–170).

Figure 1. CKD progression phenotypes observed upon increased kidney load in systemic *Cdkal1* KO mice. (L) Urine albumin levels normalized by urine creatinine (Cr) 50-week-old mice. $n = 6$. Data are presented as mean \pm SEM. $*P = 0.0152$ by Mann–Whitney U test.

Expanded View Figure 3. Validation, localization, and renal phenotype of CDKAL1 in mouse kidneys. (C) Urine albumin levels normalized by urine creatinine (Cr) 50-week-old mice. $n = 6$. Data are presented as mean \pm SEM. $*P = 0.0152$ by Mann–Whitney U test.

2) In Figure 2, I would recommend examining a few IF or IHC of podocyte specific proteins to demonstrate the molecular pathology of renal glomeruli in podocyte-specific KO mice.

Thank you for this comment. We performed immunofluorescence using antibodies against the podocyte-specific proteins nephrin and synaptopodin in the glomeruli of podocyte-specific *Cdkal1* KO mice. However, we did not observe a clear difference in the localization of nephrin and synaptopodin (Appendix Fig. S2, as shown on the next page). In contrast, podocyte-specific *Cdkal1*

KO mice displayed albuminuria and segmental sclerosis within the glomeruli (Fig. 2E, G), which are hallmarks of glomerular damage and CKD progression. We apologize that using the tissues, we could only show functional and cellular pathology and not molecular pathology due to difficulties in preparing a sufficient number of mice within the limited revision period and difficulties in separating podocytes from other cell types in the kidney tissue.

Appendix Figure S2. Reduction of podocyte-specific proteins in immunofluorescence staining of podocyte-specific *Cdkal1* KO mice. (A) Double immunofluorescence staining for nephrin, synaptopodin (green) with CDKAL1 (magenta) in the glomeruli of 20-week-old male mice. Nephrin and synaptopodin are the podocyte markers. Scale bars, 20 μm.

3) In Figure 3 and 5, F-actin staining needs to be quantified. Showing only a few representative images is sufficient.

We thank the reviewer for the helpful comment and have quantified the F-actin staining exemplified in Figs. 3I and 5C. The quantification has now been incorporated as the new Figs. 3J, K, and 5D, respectively. The quantification method has been added to the Methods section (page 23, lines: 551–555).

Figure 3. *Cdkal1* KO leads to impaired lysine codon translation, filtration barrier dysfunction, and podocyte motility defect. (J, K) Quantification of the cellular area occupied by F-actin. J: E11, k: SVI. $n = 10$; data are presented as the mean \pm SEM; **** $P < 0.0001$ by one-way ANOVA, followed by Dunnett's correction.

Figure 5. *Cd2ap* overexpression partially restores filtration barrier function and migration in *Cdkal1* KO podocytes. (D) The cellular area occupied by F-actin was quantified *Cdkal1* KO SVI cells overexpressing *Cd2ap*. $n = 10$; data are presented as the mean \pm SEM; * $P = 0.0395$ (KO1 Mock vs OE); **** $P < 0.0001$ (KO2 Mock vs OE); ** $P = 0.0011$ (sgControl Mock vs KO1 Mock) and **** $P < 0.0001$ (sgControl Mock vs KO2 Mock) by 2-way ANOVA, followed by Dunnett's correction.

4) In Figure 4, the authors only examined and confirmed CD2AP expression was reduced by KO of CDKAL1, while in Figure 4g, a list of proteins are rich in Lysine. Can the authors examine a few more of these proteins to strengthen the conclusion?

We thank the reviewer for this constructive suggestion. To address this point, among other lysine-rich proteins than CD2AP, we have chosen anillin and podocin (encoded by *NPHS2*) proteins as subjects for western blot analysis, after prioritizing proteins that 1) exhibit high podocyte specificity, 2) for which reliable commercial antibodies are available, and 3) play functionally and

pathologically important roles in podocytes. Western blot analysis revealed that the protein expression levels of Anillin and Podocin were both reduced in *Cdkal1* KO podocytes compared to those in controls. Related descriptions were added to the Results section (page 13, lines 285–288) and the figure to the new Appendix Fig. S6D–F.

Appendix Figure S6. Post-transcriptional dysregulation of podocyte structural proteins in *Cdkal1* KO podocytes and glomeruli. (D) Representative western blot images of anillin and podocin (encoded by *NPHS2*), and lysine-rich protein in *Cdkal1* KO and control SVI podocytes. (E, F) Quantification of anillin (E) and podocin (F) levels normalized to GAPDH in *Cdkal1* KO and control SVI podocytes. $n = 6-8$ each. Data are presented as the mean \pm SEM. **** $P < 0.0001$, ** $P < 0.01$, and * $P < 0.05$ by one-way ANOVA, followed by Dunnett’s correction.

Reference

Lehtonen S, Zhao F, Lehtonen E (2002) CD2-associated protein directly interacts with the actin cytoskeleton. *Am J Physiol Renal Physiol* 283: F734-743

Nakabayashi H, Taketa K, Miyano K, Yamane T, Sato J (1982) Growth of human hepatoma cells lines with differentiated functions in chemically defined medium. *Cancer Res* 42: 3858-3863

Tang VW, Briehar WM (2013) FSGS3/CD2AP is a barbed-end capping protein that stabilizes actin and strengthens adherens junctions. *J Cell Biol* 203: 815-833

Dear Prof. Tomizawa,

Thank you for submitting a revised version of your manuscript. Your study has now been seen by one of the original referees, who finds that their previous concerns have been addressed and now recommends publication of the manuscript. There remain only a few mainly editorial points that have to be addressed before I can extend formal acceptance of the manuscript:

- Please label the corresponding author in author list on the title page of manuscript.
- On the abstract page of the manuscript, please include 4-5 general keyword terms to enhance searchability.
- Please rename the "Data Statement" section to "Data Availability"
- Please rename the "Conflict of Interest" section into "Disclosure and Competing Interests Statement", in accordance with our updated Guide to Authors (<https://link.springer.com/partners/embo-press/editorial-policies#Competing%20interest%20disclosures>)
- As we are switching from a free-text author contribution statement towards a more formal statement based on Contributor Role Taxonomy (CRediT) terms, please remove the present Author Contribution section and instead specify each author's contribution(s) directly in the Author Information page of our submission system during upload of the final manuscript. See <https://casrai.org/credit/> for more information.
- FIGURE CALLOUTS: missing callouts for EV figures; callout for Appendix Table 1 should be corrected to Appendix Table S1
- Please rename the EV figures to Figure EV1-EV5 instead of Expanded View Figure 1-6
- Please make sure that the aspect ratio of the synopsis image conforms to our website's format - it should be exactly 550 pixels wide and between 300-600 pixels high.
- Figure Legends (main + EV):
 1. Please note that the legend for figure 4 is not provided in the sequential manner. This needs to be rectified.
 2. Please note that the exact p values are not provided in the legends of figures 3C, F, H, J, K; 4D, F, J; 5D, G.
 3. Please note that information related to n is missing in the legends of figures EV5 A-C
 4. Please note that the error bars are not defined in the legends of figures 2A, 4D-F
- Sections need to be named and the order should be corrected: Title page - Abstract - Keywords - Introduction - Results - Discussion - Methods - Data Availability - Acknowledgements - Disclosure and Competing Interests Statement - References - Figure Legends - Table(s) - Expanded View Figure Legends.

With best regards,
Cornelius Schneider

Cornelius Schneider, PhD
Editor | The EMBO Journal
c.schneider@embojournal.org

Please refer to our figure preparation guideline in order to ensure proper formatting and readability in print as well as on screen:

<https://link.springer.com/journal/44318/submission-guidelines#cms-Figure-and-data-presentation>

Use the link below to submit your revision:

Referee #1:

The authors have adequately addressed my issues and those of the other reviewers, and have improved the manuscript. I do not have additional comments.

All minor editorial requests have been addressed by the authors.

Dear Prof. Tomizawa,

I am pleased to inform you that your manuscript has been accepted for publication in the EMBO Journal.

You may qualify for financial assistance for your publication charges - either via a Springer Nature fully open access agreement or an EMBO initiative. Check your eligibility: <https://link.springer.com/journal/44318/how-to-publish-with-us>

Yours sincerely,

Cornelius Schneider, PhD
Editor
The EMBO Journal
c.schneider@embojournal.org

Please note that it is The EMBO Journal policy for the transcript of the editorial process (containing referee reports and your response letters) to be published as an online supplement to each paper. If you should prefer removal of any referee-only figures included in the point-by-point response(s), e.g. because they may still be used for future publication or because they have been reproduced from published work by others, please do let us know immediately via response email.

More information is available here: <https://link.springer.com/partners/embo-press/editorial-policies#Peer%20review>